# Ubiquitination-activated TAB–TAK1–IKK–NF-κB axis modulates gene expression for cell survival in the lysosomal damage response

Akinori Endo[1]\*, Chikage Takahashi[1], Naoko Ishibashi[1], Yasumasa Nishito[2], Koji Yamano[3], Keiji Tanaka[1], Yukiko Yoshida[1]\*

[1]Laboratory of Protein Metabolism, Tokyo Metropolitan Institute of Medical Science, Tokyo, Japan; [2]Technology Research Division, Tokyo Metropolitan Institute of Medical Science, Tokyo, Japan; [3]Intracellular Quality Control Project, Tokyo Metropolitan Institute of Medical Science, Tokyo, Japan

**\*For correspondence:**
endo-ak@igakuken.or.jp (AE);
yoshida-yk@igakuken.or.jp (YY)

**Competing interest:** The authors declare that no competing interests exist.

## eLife Assessment

This study presents the **important** finding that lysosomal damage triggers inflammatory signaling through ubiquitination and the TAB-TAK1-IKK-NF-kB axis. The data obtained from the unbiased transcriptomic and proteomic analyses are **convincing** and provide invaluable information to the field. Although further experiments will be required to clarify how TAB2/3 are recruited after various types of lysosome damage, this work will be of interest to researchers in the fields of organelle biology and inflammation.

**Abstract** The lysosomal damage response is important for the maintenance of cellular homeostasis in human cells. Although the mechanisms underlying the repair and autophagic elimination of damaged lysosomes have been elucidated, the early signal transduction pathways and genes induced in response to lysosomal damage remain elusive. We performed transcriptome and proteome analyses and found that the TAB–TAK1–IKK–NF-κB axis is activated by K63-linked ubiquitin chains that accumulate on damaged lysosomes. This activates the expression of various transcription factors and cytokines that promote anti-apoptosis and intercellular signaling. The findings highlight the crucial role of ubiquitin-regulated signal transduction and gene expression in cell survival and cell–cell communication in response to lysosomal damage. The results suggest that the ubiquitin system is not only involved in the removal of damaged lysosomes by lysophagy, but also functions in the activation of cellular signaling for cell survival.

## Introduction

Lysosomes are the primary degradative organelles and play a pivotal role in maintaining cellular and tissue homeostasis (*Ballabio and Bonifacino, 2020*; *Yang and Wang, 2021*). Lysosomal dysfunction is associated with various diseases, underscoring the crucial role of lysosomal integrity and its quality control in human health (*Tan and Finkel, 2023*; *Settembre and Perera, 2024*; *Bonam et al., 2019*). Lysosomes are susceptible to damage by many sources, both internal and external to the cell, which results in the permeabilization and rupture of lysosomal membranes. Crystals, amyloids, pathogens, lipids, lysosomal toxic drugs, and reactive oxygen species are among the stressors that can damage lysosomal membranes (*Papadopoulos and Meyer, 2017*; *Cantuti-Castelvetri et al., 2018*; *Mossman*

*and Churg, 1998*; *Gómez-Sintes et al., 2016*; *Wang et al., 2018*). The release of enzymes and protons from the lumen of damaged lysosomes induces oxidative stress, inflammation, and cell death, which is deleterious to cells and tissues (*Wang et al., 2018*; *Hornung et al., 2008*). The accumulation of damaged lysosomes is thus an important factor involved in the development of aging-related diseases and other conditions such as crystalline nephropathy, neurodegenerative, and infectious diseases (*Tan and Finkel, 2023*; *Settembre and Perera, 2024*; *Bonam et al., 2019*; *Nakamura et al., 2020*).

To overcome lysosomal damage and restore lysosomal functionality, cells have developed a set of response pathways that is collectively referred to as the lysosomal damage response (*Papado-poulos and Meyer, 2017*; *Meyer and Kravic, 2024*). It has been reported that signal mediators, such as AMP-activated protein kinase (AMPK) and mammalian target of rapamycin (mTOR), exert the signal transduction in the lysosomal damage response (*Jia et al., 2018*; *Jia et al., 2020a*). The quality control mechanisms for damaged lysosomes include repair, elimination, and regeneration. Lysosomal membrane damage triggers the recruitment of the endosomal sorting complex required for transport (ESCRT) to repair the damaged membrane (*Meyer and Kravic, 2024*; *Radulovic et al., 2018*). Current studies have shown that endoplasmic reticulum (ER)-lysosome lipid transfer is crucial for lysosomal membrane repair (*Wang et al., 2025*; *Radulovic et al., 2022*). Furthermore, stress granules (SGs) have been demonstrated to play a pivotal role in the repair process (*Bussi et al., 2023*). When the damage is extensive and repair is not feasible, the damaged lysosome under-goes autophagic elimination in a process known as lysophagy (*Meyer and Kravic, 2024*; *Papa-dopoulos et al., 2020*). In lysophagy, lysosomal membrane permeabilization leads to extensive ubiquitination of exposed proteins. This triggers the recruitment of ubiquitin-binding autophagy receptors, which activates autophagy to sequester and degrade damaged lysosomes (*Meyer and Kravic, 2024*; *Eapen et al., 2021*). Protein ubiquitination is an important form of posttranslational modification that is comparable to protein phosphorylation. Protein ubiquitination in lysophagy occurs through the formation of Lys-63-linked ubiquitin chains (K63 ubiquitin chains) (*Gahlot et al., 2024*). Substrate ubiquitination in damaged lysosomes is catalyzed by the E2 ubiquitin-conjugating enzyme UBE2QL1 and several ubiquitin E3 ligases, including TRIM16, CUL1–SKP1–FBXO27, and CUL4A–DDB1–WDFY1 (*Chauhan et al., 2016*; *Koerver et al., 2019*; *Yoshida et al., 2017*; *Teranishi et al., 2022*).

Transcription factor EB (TFEB), a master regulator of lysosome- and autophagy-related gene expression, is activated during the repair and elimination of damaged lysosomes, which results in the regeneration of lysosomes as part of the lysosomal damage response (*Sardiello et al., 2009*; *Puer-tollano et al., 2018*; *Shariq et al., 2024*). Under unstimulated conditions, TFEB is phosphorylated by lysosome-associated mTORC1 and binds to 14–3–3 proteins, resulting in the cytoplasmic retention of TFEB (*Settembre et al., 2013*). However, lysosomal permeabilization causes the dissociation of mTORC1 from lysosomes, which leads to the dephosphorylation of TFEB and its translocation to the nucleus to activate the expression of lysosome- and autophagy-related genes (*Tan and Finkel, 2023*; *Jia et al., 2018*; *Settembre et al., 2013*).

The elimination of damaged lysosomes through lysophagy requires a period of more than half a day, during which the cell is vulnerable (*Maejima et al., 2013*). It is therefore assumed that there are cellular mechanisms that cooperate with lysophagy to maintain cellular health in the lysosomal damage response. In previous work, we demonstrated that K63 ubiquitin chains, which accumulate on dysfunctional endosomes, induce cytokine production by modulating gene expression in a TGF-beta-activated kinase 1 (TAK1) and TAK1-binding protein (TAB) dependent manner (*Endo et al., 2024a*; *Endo et al., 2024b*). K63 ubiquitin chains also accumulate in damaged lysosomes, which serve as an initiating signal for lysophagy. This prompted us to investigate whether gene expression is induced by ubiquitin-mediated activation of the TAB–TAK1 pathway in the lysosomal damage response, as observed in the endosomal stress response. The transcriptome of cells exposed to lysosomal damaging agents for 30 min has been described (*Jia et al., 2022*); however, genome-wide gene expression patterns associated with the lysosomal damage response remain unclear. In this study, we performed transcriptome and proteome analyses to comprehensively examine the regulation of gene expression in the lysosomal damage response and to identify the underlying signal transduction pathways. We found that accumulation of K63 ubiquitin chains in damaged lysosomes activates the TAB–TAK1–IKK–NF-κB pathway, which in turn induces the expression of various transcription factors and cytokines. We elucidated the critical role of the TAB–TAK1–IKK–NF-κB pathway in cell survival

and intercellular signal transduction. Collectively, the findings suggest the existence of a universal response mechanism against the intracellular accumulation of K63 ubiquitin chains.

## Results

### Lysosomal damage has a global impact on the transcriptome and proteome

To comprehensively investigate the changes in gene expression caused by lysosomal damage, we performed transcriptome analysis and compared mRNA expression between cells treated with L-leucyl–L-leucine methyl ester (LLOMe), a well-characterized inducer of lysosomal damage (*Uchimoto et al., 1999*), and control cells. We used RPE-1 cells, which are commonly used in aging research. Transcriptome data revealed substantial alterations in gene expression in response to lysosomal damage. We identified >1000 genes (946 upregulated and 164 downregulated genes from >13,000 transcripts with $Log_2$ fold change (FC) >|1| and p<0.05) that exhibited significant changes at 2 hr after LLOMe treatment (*Figure 1A*). Proteome analysis was performed to determine the correlation between proteomic changes and the observed alterations in the transcriptome. The analysis identified 86 upregulated and 60 downregulated proteins (of >10,000 proteins with $Log_2$ FC >|1| and p<0.05) (*Figure 1B*). There was a modest correlation between the significant upregulation at the RNA level and that at the protein level, with a correlation coefficient (*R*) of 0.5899 from 302 data points (*Figure 1C*, *Figure 1—figure supplement 1A*). The representative targets interleukin-1 β (IL1β), IL6, interferon regulatory factor 1 (IRF1), and the proto-oncogenes c-Jun (JUN) and c-Fos (FOS) were upregulated at both the RNA and protein levels (*Figure 1C*, *Figure 1—figure supplement 1A and B*). This indicates that alterations in the transcriptome were, at least in part, responsible for those observed in the proteome. Genes showing significant upregulation in the transcriptome and proteome analyses were subjected to gene enrichment analysis using the Gene Ontology Molecular Function reference dataset (*Figure 1D and E*). The results showed that these genes were primarily enriched in transcription factors and cytokines (labeled in red and blue, respectively, in *Figure 1D*). Analysis using a Molecular Signatures Database hallmark gene set (*Liberzon et al., 2015*), a phenotypic reference dataset, indicated that genes upregulated at both the RNA and protein levels were involved in biological regulatory processes, including the inflammatory response and apoptosis (*Figure 1E*). It is noteworthy that the transcription factor prediction using the DoRothEA regulon (*Garcia-Alonso et al., 2019*; *Badia-I-Mompel et al., 2022*) identified the p65 (RELA) and p105 (NFKB1) subunits of the NF-κB family as the primary transcription factors regulating gene expression in the lysosomal damage response (*Figure 1F*). This suggests that NF-κB activation is involved in the regulation of gene expression in the lysosomal damage response. The upregulation of IL1β, IL6, IRF1, JUN, and FOS at the protein level was confirmed by immunoblotting (*Figure 1G*).

Taken together, the results suggest that in the lysosomal damage response, the proteome is remodeled according to alterations in the transcriptome; this remodeling is mediated primarily through NF-κB and involves many transcription factors and cytokines, which may potentially lead to the regulation of inflammation and apoptosis (*Figure 1H*).

### Lysosomal damage activates TAK1 in a ubiquitin- and TAB-dependent manner

Lysosomal membrane damage exposes glycoproteins, such as LAMP2. Cytosolic galectin proteins, including Galectin-3 (Gal-3), are recruited to the lysosomal membrane by recognizing beta-galactoside-containing glycans on exposed proteins. These exposed lysosomal proteins are extensively ubiquitinated by ubiquitin ligases, resulting in the accumulation of ubiquitinated proteins within damaged lysosomes (*Yoshida et al., 2017*; *Teranishi et al., 2022*; *Jia et al., 2020b*). An endosomal stress response mechanism was recently described, whereby K63 ubiquitin chains accumulate on dysfunctional endosomes, activating TAK1 in a TAB-dependent manner (*Endo et al., 2024a*). TAB2 and TAB3 (a paralog of TAB2) function as ubiquitin decoders that specifically recognize K63 ubiquitin chains via the C-terminal Npl4 zinc finger (NZF) domain (*Kulathu et al., 2009*; *Kanayama et al., 2004*). The finding that TAB binding to unanchored K63 ubiquitin chains is sufficient to activate TAK1 in vitro (*Xia et al., 2009*) suggests that the conjugation of K63 ubiquitin chains to bulky substrates on endosomes is sufficient to facilitate TAB-dependent TAK1 activation. This prompted us to investigate whether the

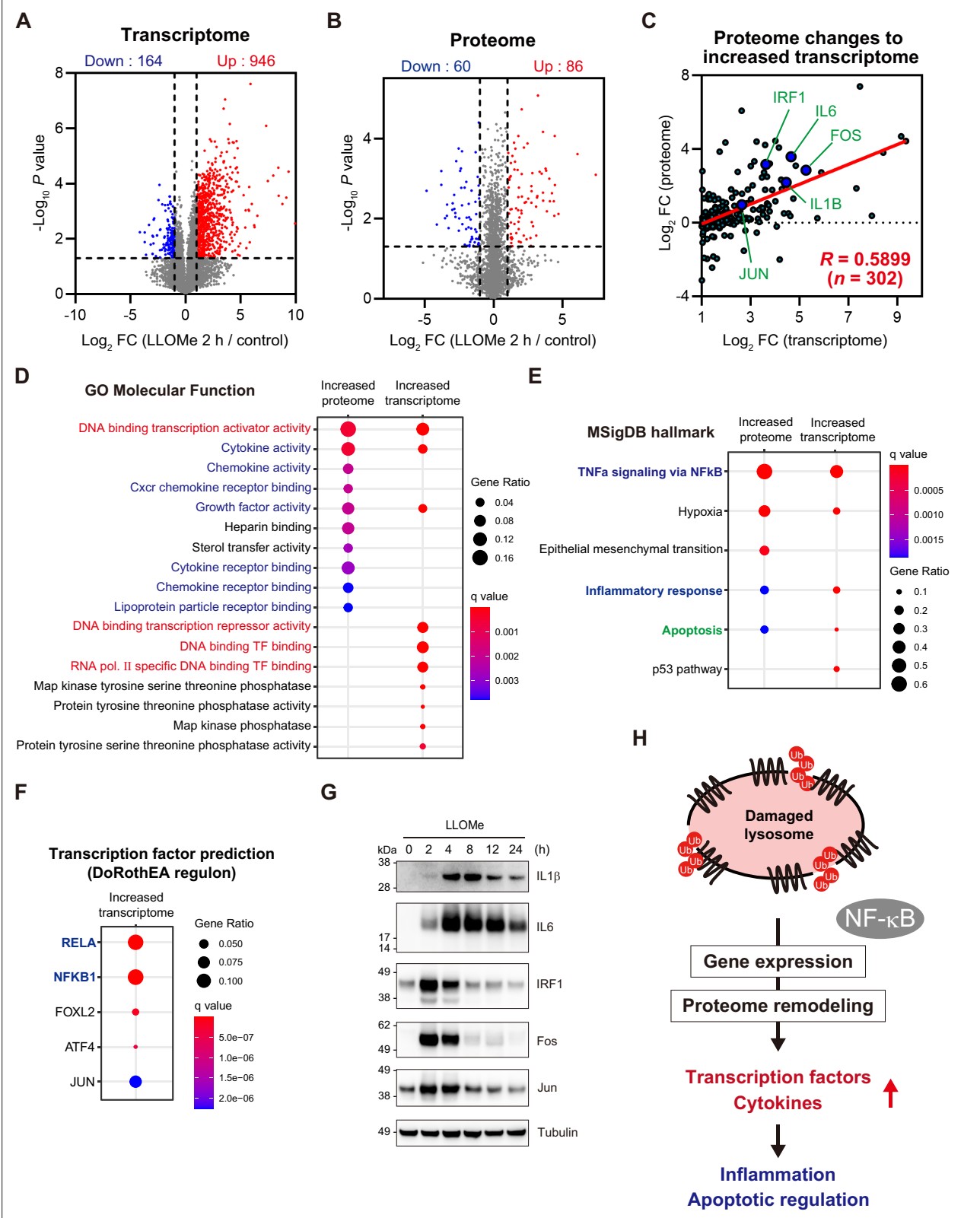

**Figure 1.** Lysosomal damage has a global impact on the transcriptome and proteome. (**A, B**) The mean $Log_2$ fold change (FC, LLOMe 2 hr/control) and $-Log_{10}$ p-value of the transcriptome (**A**) and proteome (**B**) in RPE-1 cells are indicated on the x and y axes, respectively. Genes significantly upregulated or downregulated are labeled in red and blue, respectively. (**C**) For genes showing significant upregulation in the transcriptome ($Log_2$ FC >1 and p<0.05), the correlation between the mean $Log_2$ FC of the transcriptome (from A, x axis) and the proteome (from B, y axis) is shown with a coefficient of

*Figure 1 continued on next page*

*Figure 1 continued*

correlation (*R*=0.5899, *n*=302). (**D, E**) The bubble plots show the outcomes of gene enrichment analyses based on Gene Ontology Molecular Function (GOMF) (**D**) and Molecular Signatures Database (MSigDB) hallmark gene sets (**E**) for genes upregulated at the protein and RNA levels (top ten and five categories for GOMF and MSigDB hallmark gene sets, respectively). The color and size of the bubbles indicate the q value and gene ratio, respectively. The categories related to transcription, cytokine/growth factor, and apoptosis are labeled in red, blue, and green, respectively. (**F**) The bubble plot illustrates the DoRothEA regulon-based prediction of transcription factors responsible for the induction of genes upregulated in cells treated with LLOMe (top five transcription factors). The color and size of the bubbles indicate the q value and gene ratio, respectively. NF-κB components are labeled in blue. (**G**) RPE-1 cells were treated with LLOMe for 2 hr and then washed. The total incubation times are indicated. Total cell lysates were subjected to immunoblotting with the indicated antibodies. (**H**) Schematic model of the cellular response to lysosomal damage.

The online version of this article includes the following source data and figure supplement(s) for figure 1:

**Source data 1.** Original files for western blot analysis for *Figure 1G*.

**Source data 2.** Uncropped full images displayed in *Figure 1G*.

**Figure supplement 1.** Representative targets induced by lysosomal damage.

accumulation of K63 ubiquitin chains in other regions of the cytoplasm also results in TAK1 activation. TAK1 controls the function of the p62/sequestosome, an autophagy receptor, and TAB2 undergoes TRIM38-mediated lysosomal degradation in response to lysosomal damage (*Hu et al., 2014*; *Kehl et al., 2019*). However, whether TAK1 is activated in a TAB-dependent manner and the role of the TAB–TAK1 signaling pathway in the lysosomal damage response remain unanswered questions.

To determine whether the TAB–TAK1 signaling pathway is activated in response to lysosomal damage, we initially examined the subcellular localization of TAB2 and TAK1. The recruitment of Gal-3 and K63 ubiquitin chains to damaged lysosomes was confirmed in RPE-1 cells (*Figure 2—figure supplement 1A*). The co-localization of TAB2 and TAK1 with K63 ubiquitin chains at damaged lysosomes was observed 5 min after LLOMe treatment (*Figure 2A*). The signals of LAMP1, a lysosomal marker protein, were enhanced, and LAMP1-positive puncta were increased in size in RPE-1 cells treated with LLOMe. It is plausible that this phenomenon reflects the morphological alterations in lysosomes under our experimental conditions, possibly due to lysosomal membrane damage (*Figure 2A*, *Figure 2—figure supplement 1A*). Immunoblotting analysis showed that the accumulation of K63 ubiquitin chains in the cell was accompanied by an increase in active phosphorylated TAK1, which was first detected 5 min after LLOMe treatment (*Figure 2B*). TFEB isolated from cells treated with LLOMe exhibited a downshift on the gel, which is a characteristic of TFEB activation (*Ogura et al., 2022*). This indicates that a lysosomal damage response was activated under these experimental conditions (*Figure 2B*). TAK1 activation was abolished by the E1 ubiquitin-activating enzyme inhibitor TAK-243 and depletion of TAB2 and TAB3 (*Figure 2C and D*). These results suggest that TAK1 is activated in a ubiquitin- and TAB-dependent manner in the lysosomal damage response. To further explore the potential involvement of K63 ubiquitination in TAK1 activation, we performed an add-back experiment using siRNA-resistant TAB2 WT and mutants incapable of binding to K63 ubiquitin chains, dNZF and E685A. The NZF domain of TAB2 is well characterized to specifically associate with K63 ubiquitin chains (*Kulathu et al., 2009*). We investigated whether the add-back of TAB2 mutants rescues the activation of TAK1 in TAB2-depleted cells (*Figure 2E*). TAB2 WT, but not dNZF and E685A, restored TAK1 activation in response to lysosomal damage, suggesting that the specific interaction of TAB proteins and K63 ubiquitin chains is a key mechanism to activate TAK1. Unexpectedly, the treatment of an E1 inhibitor TAK-243 that abolished the lysosomal accumulation of K63 ubiquitin chains did not inhibit the recruitment of TAB2 to damaged lysosomes, suggesting that the recruitment of TAB proteins to damaged lysosomes is independent of the association with K63 ubiquitin chains (*Figure 2—figure supplement 1B*). Collectively, it is postulated that TAB proteins require the interaction with K63 ubiquitin chains for TAK1 activation, but not for recruitment to damaged lysosomes. We next examined whether other lysosomal damage inducers, glycyl-L-phenylalanine 2-naphthylamide (GPN) and DC661, also activate TAK1. It has been shown that GPN and DC661 induce lysosomal membrane permeabilization through hyperosmotic stress and lipid peroxidation in lysosomes, respectively (*Scott et al., 2025*; *Bhardwaj et al., 2023*). In cells treated with GPN and DC661, K63 ubiquitin chains were accumulated, possibly due to a defect in the endo-lysosome machinery (*Figure 2—figure supplement 1C and D*). However, the lysosomal accumulation of Gal-3 was significantly lower than that observed in cells treated with LLOMe, suggesting that GPN and DC661 did not induce severe lysosomal membrane permeabilization (*Figure 2—figure supplement 1C and D*). Under these conditions, the phosphorylation of TAK1

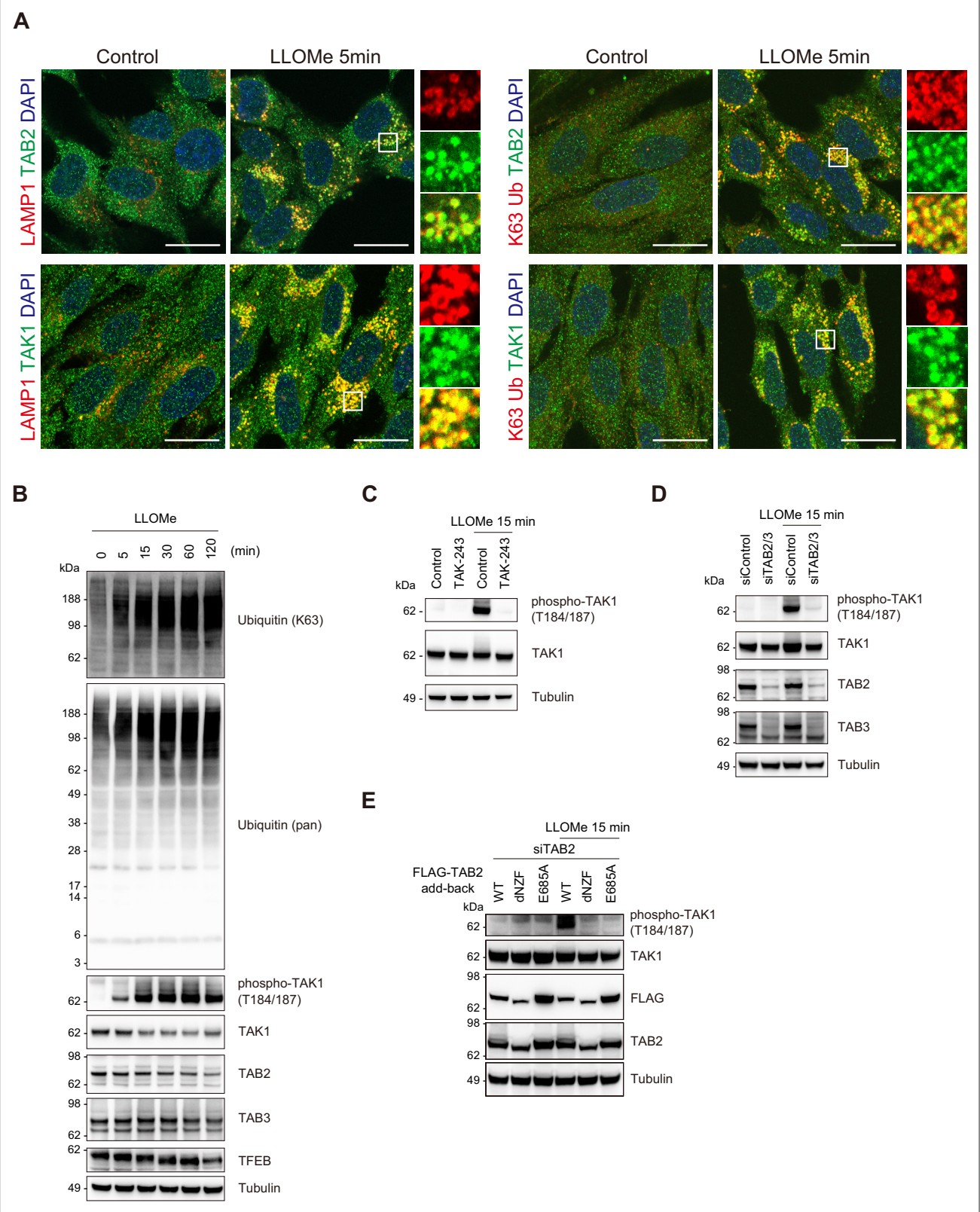

**Figure 2.** Lysosomal damage activates TAK1 in a ubiquitin- and TAB-dependent manner. (**A**) RPE-1 cells treated with L-leucyl–L-leucine methyl ester (LLOMe) for 5 min were immunostained with the indicated antibodies and DAPI. Scale bar, 20 μm. (**B**) Total cell lysates from RPE-1 cells treated with LLOMe for the indicated times were subjected to immunoblotting with the indicated antibodies. (**C**) Total cell lysates from RPE-1 cells pre-treated with TAK-243 for 15 min and treated with LLOMe for 15 min were subjected to immunoblotting with the indicated antibodies. (**D**) Total cell lysates from

*Figure 2 continued on next page*

*Figure 2 continued*

RPE-1 cells transfected with the indicated siRNAs and treated with LLOMe for 15 min were subjected to immunoblotting with the indicated antibodies. (**E**) HeLa cells stably expressing FLAG-TAB2 WT, dNZF, and E685A were transfected with the indicated siRNAs and treated with LLOMe for 15 min. Total cell lysates were subjected to immunoblotting with the indicated antibodies.

The online version of this article includes the following source data and figure supplement(s) for figure 2:

**Source data 1.** Original files for western blot analysis for *Figure 2B and C*.

**Source data 2.** Uncropped full images displayed in *Figure 2B and C*.

**Source data 3.** Original files for western blot analysis for *Figure 2D and E*.

**Source data 4.** Uncropped full images displayed in *Figure 2D and E*.

**Figure supplement 1.** The analysis in various lysosome-damaging conditions.

**Figure supplement 1—source data 1.** Original files for western blot analysis for *Figure 2—figure supplement 1E*.

**Figure supplement 1—source data 2.** Uncropped full images displayed in *Figure 2—figure supplement 1E*.

was not observed (*Figure 2—figure supplement 1E*). These findings indicate that the exposure of K63 ubiquitin chains within the cytosol following severe lysosomal membrane permeabilization functions as an initial trigger for TAB-mediated TAK1 activation in the lysosomal damage response.

## The TAB–TAK1 pathway plays a pivotal role in the induction of cytokines and transcription factors in the lysosomal damage response

To gain further insight into the involvement of the TAB–TAK1 pathway in the regulation of gene expression associated with the lysosomal damage response, an additional transcriptome analysis was performed on TAB- and TAK1-depleted cells treated with LLOMe for 2 hr. The genes upregulated in response to lysosomal damage were classified into six distinct clusters within the dataset (*Figure 3A*). Clusters 4 and 5 comprised genes expressed in a TAB- and TAK1-dependent manner (*Figure 3A*). Most of the genes upregulated in a TAB- and TAK1-dependent manner showed overlapping expression patterns (*Figure 3—figure supplement 1A and B*); these genes were enriched for cytokines and growth factors, and their functions were linked to inflammatory responses and apoptosis, as shown in the gene enrichment analysis (*Figure 3—figure supplement 1C and D*). Transcription factor prediction based on the DoRothEA regulon indicated that genes in cluster 4, whose expression was inhibited by depletion of TAB and TAK1, were targets of the NF-κB family, including RelA, NFKB1, and c-Rel (REL) (*Figure 3B*). The NF-κB family members were identified as the primary transcription factors regulating gene expression in the lysosomal damage response (*Figure 1F*). This indicates that the TAB–TAK1 pathway serves as a primary axis for the activation of NF-κB in the lysosomal damage response, thereby inducing target gene expression. Cytokines and growth factors were enriched in cluster 4, and transcription factors were enriched in cluster 5, in which gene expression was partially suppressed (*Figure 3C and E*). Regulators of the inflammatory response and apoptotic functions were commonly enriched in clusters 4 and 5 (*Figure 3D*). The results of immunoblotting showed that the protein expression of IL1β, IL6, IRF1, homeobox protein Nkx-3.1 (NKX3.1), and JUN was consistent with the TAB- and TAK1-mediated gene expression. Conversely, depletion of TAB and TAK1 did not affect the expression of FOS at both the mRNA and protein levels (*Figure 3F*, *Figure 3—figure supplement 1E*). Furthermore, the downshift of TFEB, which indicates the activation of TFEB, was not affected by depletion of TAB and TAK1 (*Figure 3F*). This suggests that TFEB activation is independent of the TAB–TAK1 pathway in the lysosomal damage response.

In conclusion, transcriptome analysis in TAB- and TAK1-depleted cells indicates that the TAB–TAK1 pathway induces the expression of cytokines and transcription factors, potentially via NF-κB, in the lysosomal damage response, thereby regulating inflammatory response and apoptosis.

## The K63 Ub–TAB–TAK1–IKK–NF-κB pathway is activated in the lysosomal damage response

To elucidate the downstream cascade of the TAB–TAK1 signaling pathway in the early response to lysosomal damage, we performed proteome and phosphoproteome analyses using the TAK1 inhibitor HS-276 (*Figure 4A*, *Figure 4—figure supplement 1A*). The phosphoproteome analysis showed that the phosphorylation levels of stress-induced mitogen-activated protein kinases (MAPKs), specifically

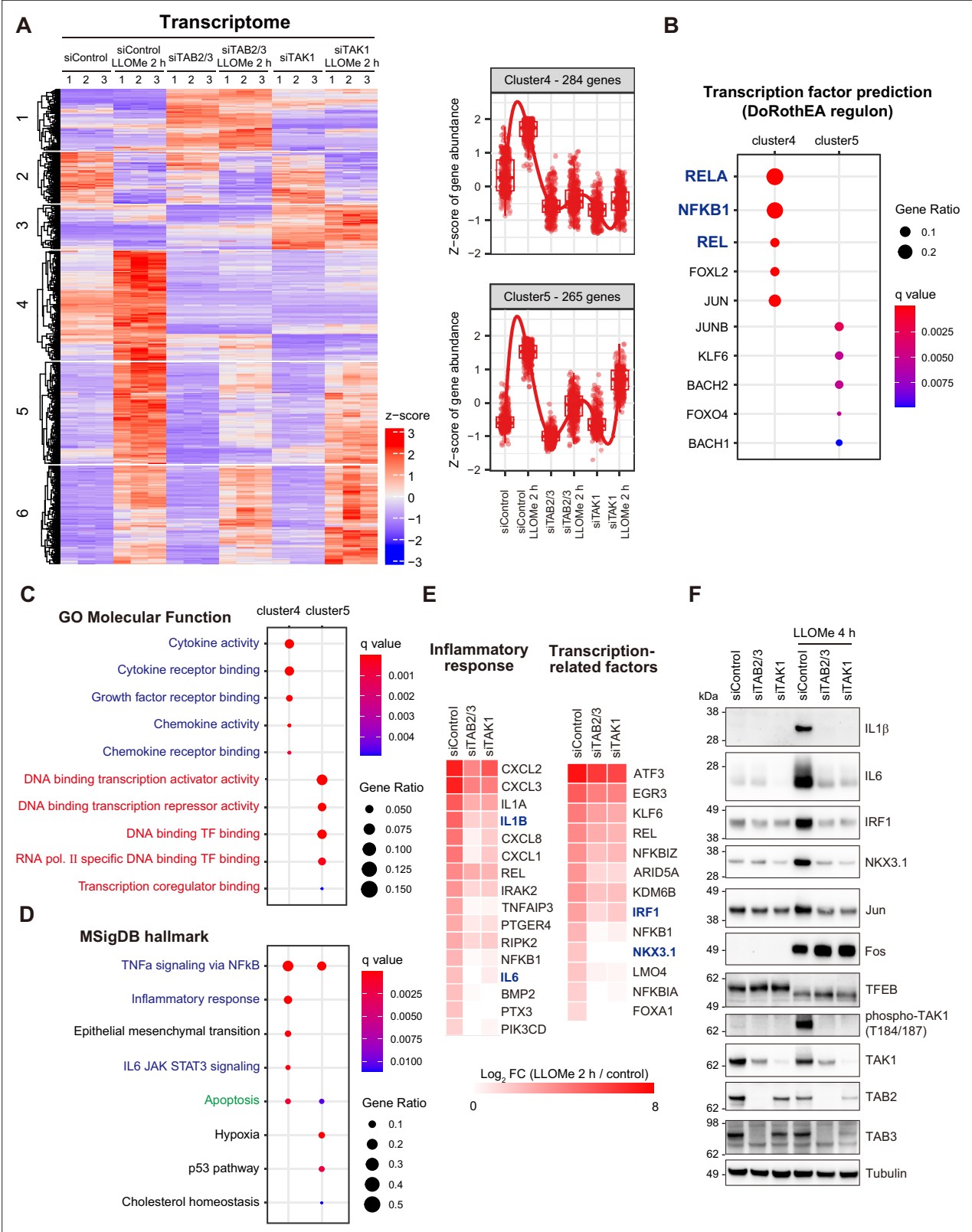

**Figure 3.** The TAB–TAK1 pathway activates cytokines and transcription factors in response to lysosomal damage. (A) Heatmap showing the changes in the transcriptome of each sample. Clusters 4 and 5 include the genes upregulated by L-leucyl–L-leucine methyl ester (LLOMe) in a TAB- and TAK1-dependent manner. (B) The bubble plot shows the DoRothEA regulon-based prediction of transcription factors responsible for the induction of genes assigned to clusters 4 and 5 in (A) (top five transcription factors). The color and size of bubbles indicate q value and gene ratio, respectively. NF-κB

*Figure 3 continued on next page*

*Figure 3 continued*

components are labeled in blue. (**C, D**) The bubble plots show the outcomes of gene enrichment analyses based on Gene Ontology Molecular Function (GOMF) (**C**) and Molecular Signatures Database (MSigDB) hallmark gene sets (**D**) for genes assigned to clusters 4 and 5 in (**A**) (top five categories). The color and size of the bubbles indicate the q value and gene ratio, respectively. The categories related to transcription, cytokine/growth factor, and apoptosis are labeled in red, blue, and green, respectively. (**E**) Heatmaps showing the genes upregulated in a TAB- and TAK1-dependent manner within the categories of inflammatory response (left) and transcription-related factors (right). The color intensity indicates the $Log_2$ FC (LLOMe 2 hr/control). (**F**) Total cell lysates from RPE-1 cells transfected with the indicated siRNAs, treated with LLOMe for 2 hr, and washed for 2 hr were subjected to immunoblotting with the indicated antibodies.

The online version of this article includes the following source data and figure supplement(s) for figure 3:

**Source data 1.** Original files for western blot analysis for *Figure 3F*.

**Source data 2.** Uncropped full images displayed in *Figure 3F*.

**Figure supplement 1.** TAB- and TAK1-dependent gene expression in the lysosomal damage response.

p38α (MAPK14, pT180/pY182) and JNK3 (MAPK10, pY223), as well as MK2 (MAPKAPK2, pT334), a downstream target of MAPKs, increased in response to lysosomal damage, and this increase was dependent on TAK1 (*Figure 4—figure supplement 1B and C*). Phosphorylation of T180/Y182 in p38α and Y223 in JNK3 activates the kinase function (*Raingeaud et al., 1996*; *Davis, 2000*). The TAK1-dependent phosphorylation of Y223 in JNK3 in the lysosomal damage response was consistent with a previous report that lysosome rupture activates the TAK1–JNK pathway in THP-1 cells (*Okada et al., 2014*). Additionally, T334 of MK2 is a primary site of phosphorylation by MAPKs (*Ben-Levy et al., 1995*). These findings suggest that lysosomal damage induces TAK1-dependent activation of MAPKs. Proteome analysis showed that the inhibitor of NF-κB α (IκBα) undergoes TAK1-dependent degradation 30 min after LLOMe treatment (*Figure 4A and B*). The inhibitor of NF-κB kinase (IKK) complex, which comprises IKKα, β, and γ, functions downstream of TAK1 and phosphorylates IκBα, thereby inducing its degradation (*Wang et al., 2001*). Similarly, in response to ubiquitin-mediated endosomal stress, the NF-κB pathway is activated in an IKK-dependent manner (*Endo et al., 2024a*). The degradation of IκBα is a hallmark phenomenon upstream of NF-κB activation (*Tak and Firestein, 2001*), which is primarily triggered by IKK-mediated phosphorylation. This leads to the hypothesis that the IKK complex activated by the TAB–TAK1 pathway promotes the phosphorylation of IκBα, which leads to its degradation and NF-κB activation in the lysosomal damage response.

We therefore sought to characterize the kinase cascade that is activated in the lysosomal damage response. We confirmed the phosphorylation and subsequent degradation of IκBα (*Figure 4C*). We also found that MAPKs and IKKs were activated concomitantly with a notable phosphorylation of IKK, JNK, and p38, and a relatively minimal phosphorylation of ERK (*Figure 4C*). The increase in the phosphorylation levels of IKK, JNK, p38, and IκBα occurred in a ubiquitin-dependent manner, as the induction was abolished by TAK-243 (*Figure 4D*). Depletion of TAB, TAK1, and IKK suppressed IκBα phosphorylation, and the phosphorylation of JNK and p38 was TAB- and TAK1-dependent (*Figure 4E and F*). These findings indicate that the phosphorylation of IκBα is induced in a TAB–TAK1–IKK pathway-dependent manner. Additionally, it can be postulated that MAPKs, including JNK and p38, are activated by the TAB–TAK1 pathway.

TANK-binding kinase 1 (TBK1), a putative NF-κB regulator, has been reported to be activated in the lysosomal damage response (*Eapen et al., 2021*). We examined whether TBK1 coordinately functions with the TAB-TAK1 pathway to facilitate NF-κB activation. Consistent with the previous report, TBK1 was activated in response to LLOMe treatment (*Figure 4—figure supplement 1D*). Depletion of TAB and TAK1 led to a slight reduction in the activation of TBK1 (*Figure 4—figure supplement 1E*). The TBK1 inhibitor BX-795 did not affect TAK1 activation but abolished the phosphorylation of IKK and IκBα (*Figure 4—figure supplement 1F*). With regard to the ubiquitination in the lysosomal damage response, the linear ubiquitin chain assembly complex (LUBAC)-mediated M1 ubiquitination has been implicated in NF-κB activation (*Zein et al., 2025*). To explore the functional link between K63 and M1 ubiquitination, we examined the effects of LUBAC depletion on the TAB-TAK1 pathway and the subsequent NF-κB activation. Depletion of RNF31 (also known as HOIP), a component of LUBAC, exhibited no or minimal effects on TAK1 activation but abolished the phosphorylation of IKK and IκBα (*Figure 4—figure supplement 1G*). These findings suggest that TBK1 and LUBAC are indispensable for NF-κB activation in the lysosomal damage response.

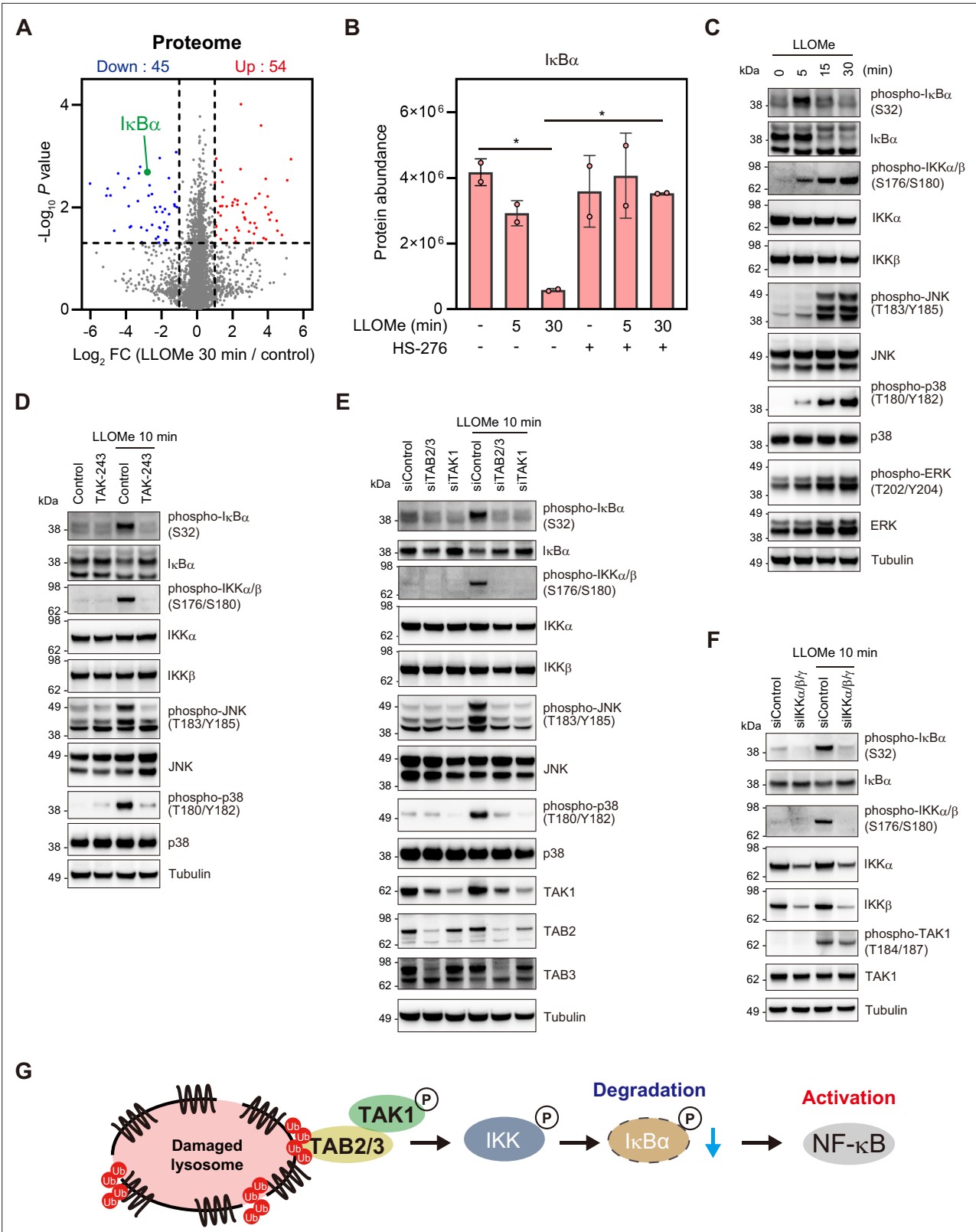

**Figure 4.** The K63 Ub–TAB–TAK1–IKK–NF-κB pathway induces the expression of cytokines and transcription factors. (**A**) The mean Log$_2$ fold change (FC) (L-leucyl–L-leucine methyl ester [LLOMe] 30 min/control) and −Log$_{10}$ p-value of the proteome are indicated on the x and y axes, respectively. Proteins significantly upregulated or downregulated are labeled in red and blue, respectively. (**B**) IκBα protein abundance determined by mass spectrometry (MS) analysis. The individual values, mean, and standard deviation (SD) of the mean of protein abundance are presented. The mean ± SD values were

*Figure 4 continued on next page*

*Figure 4 continued*

calculated from two biological replicates. *p<0.05 (one-way ANOVA with Dunnett's test). (**C**) Total cell lysates from RPE-1 cells treated with LLOMe for the indicated times were subjected to immunoblotting with the indicated antibodies. (**D**) Total cell lysates from RPE-1 cells pre-treated with TAK-243 for 15 min and treated with LLOMe for 10 min were subjected to immunoblotting with the indicated antibodies. (**E, F**) Total cell lysates from RPE-1 cells transfected with the indicated siRNAs and treated with LLOMe for 10 min were subjected to immunoblotting with the indicated antibodies. (**G**) Schematic model of the cellular signaling pathways activated in response to lysosomal damage.

The online version of this article includes the following source data and figure supplement(s) for figure 4:

**Source data 1.** Original files for western blot analysis for *Figure 4C*.

**Source data 2.** Uncropped full images displayed in *Figure 4C*.

**Source data 3.** Original files for western blot analysis for *Figure 4D*.

**Source data 4.** Uncropped full images displayed in *Figure 4D*.

**Source data 5.** Original files for western blot analysis for *Figure 4E*.

**Source data 6.** Uncropped full images displayed in *Figure 4E*.

**Source data 7.** Original files for western blot analysis for *Figure 4F*.

**Source data 8.** Uncropped full images displayed in *Figure 4F*.

**Figure supplement 1.** Signaling pathways in the lysosomal damage response.

**Figure supplement 1—source data 1.** Original files for western blot analysis for *Figure 4—figure supplement 1D and E*.

**Figure supplement 1—source data 2.** Uncropped full images displayed in *Figure 4—figure supplement 1D and E*.

**Figure supplement 1—source data 3.** Original files for western blot analysis for *Figure 4—figure supplement 1F*.

**Figure supplement 1—source data 4.** Uncropped full images displayed in *Figure 4—figure supplement 1F*.

**Figure supplement 1—source data 5.** Original files for western blot analysis for *Figure 4—figure supplement 1G*.

**Figure supplement 1—source data 6.** Uncropped full images displayed in *Figure 4—figure supplement 1G*.

The IKK complex, which is activated by the TAB–TAK1 pathway in response to lysosomal damage, has been shown to induce the phosphorylation and degradation of IκBα, potentially leading to the activation of NF-κB (*Figure 4G*). Furthermore, TBK1 and LUBAC were required for NF-κB activation, but not for TAK1 activation, indicating that TBK1 and LUBAC function downstream of or parallel to the TAB-TAK1 pathway.

## The K63 Ub–TAB–TAK1–NF-κB pathway regulates the expression of cytokines and transcription factors essential for cell survival and intercellular signaling

IL1β, IL6, IRF1, and NKX3.1 expression was associated with the lysosomal damage response and dependent on the TAB–TAK1 pathway (*Figure 3E and F*, *Figure 3—figure supplement 1E*). The expression of these genes was suppressed by treatment with TAK-243 and HS-276 (*Figure 5A and B*). Depletion of IKKs suppressed target gene expression (*Figure 5C*). These findings demonstrate that the expression of these genes is induced by the ubiquitin-dependent activation of the TAB–TAK1–IKK pathway in the lysosomal damage response.

Gene enrichment data based on the results of transcriptome and proteome analyses indicated that TAB–TAK1 pathway-dependent gene expression plays a role in the regulation of apoptosis and the inflammatory response (*Figure 3D*, *Figure 3—figure supplement 1D*). We thus examined the impact of the K63 Ub–TAB–TAK1–IKK–NF-κB pathway on these cellular functions. Lysosomal damage for 8 hr did not induce apoptosis; however, inhibition of ubiquitination by TAK-243 dramatically induced apoptosis in HeLa and RPE-1 cells (*Figure 5D and E*, *Figure 5—figure supplement 1A*). This indicates that ubiquitination is essential for cell survival in the lysosomal damage response. Similarly, inhibition of TAK1 by HS-276 and depletion of TAB, TAK1, and IKKs following lysosomal damage for 24 hr promoted apoptosis in HeLa cells (*Figure 5D–G*). TAK1 inhibition induced apoptosis in RPE-1 cells to a lesser extent compared to HeLa cells (*Figure 5D and E*, *Figure 5—figure supplement 1A*). In addition, the inhibition of JNK by JNK-IN-8 promoted apoptosis following lysosomal damage for 12 hr, which is a later time point than TAK-243 and HS-276 (*Figure 5—figure supplement 1B*). These findings demonstrate that the K63 Ub–TAB–TAK1 pathway coordinately exerts anti-apoptotic effects

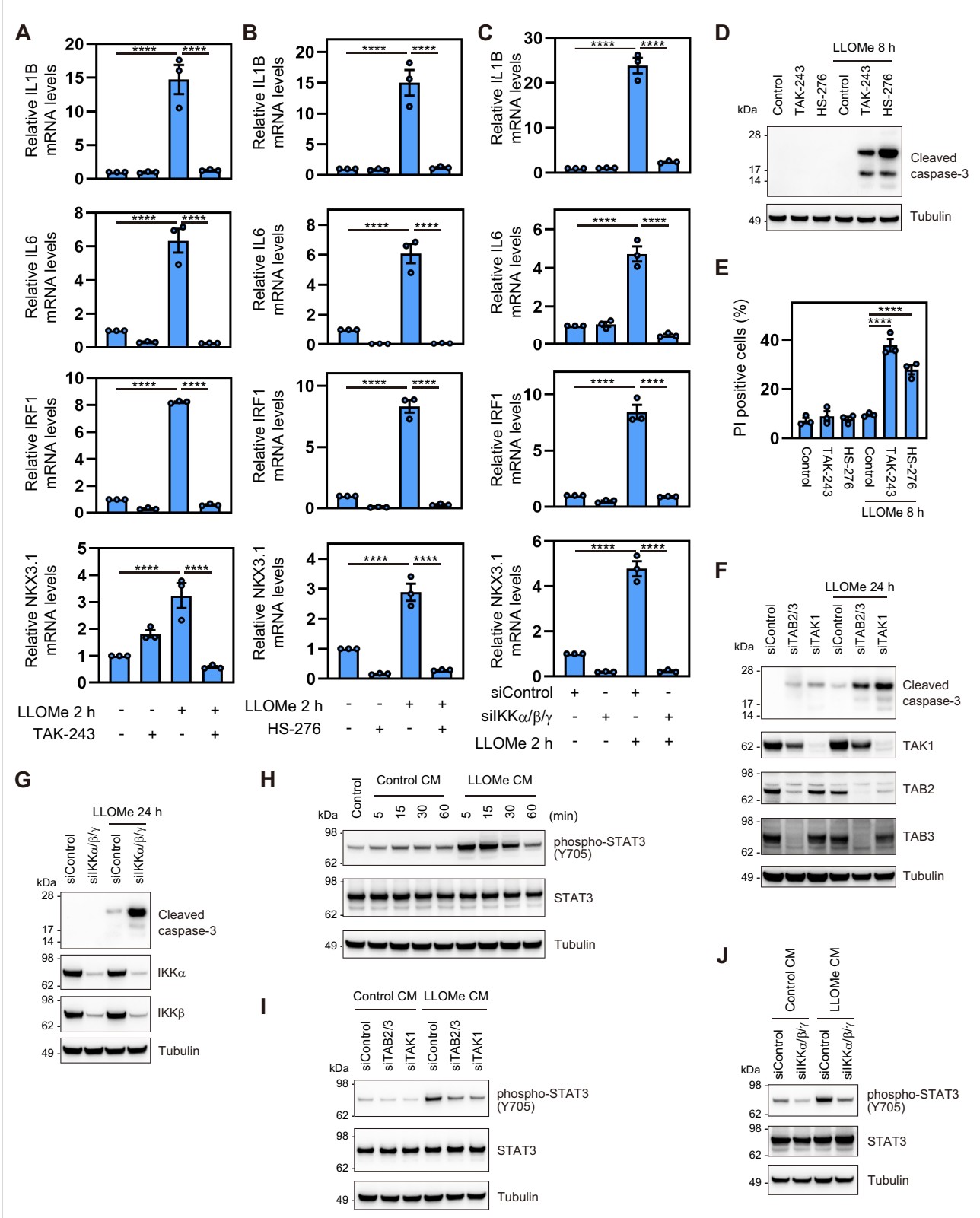

**Figure 5.** The K63 Ub–TAB–TAK1–IKK–NF-κB pathway promotes cell survival and intercellular signaling. (**A, B**) Total RNA from RPE-1 cells pre-treated with TAK-243 (**A**) and HS-276 (**B**) for 15 min and treated with L-leucyl–L-leucine methyl ester (LLOMe) for 2 hr was analyzed by RT-qPCR. Target mRNA levels were normalized to GAPDH mRNA levels; the expression levels in control cells were set to 1. The individual values, mean, and standard error of the mean (SEM) of relative mRNA levels are presented. The mean ± SEM values were calculated from three biological replicates. ****p<0.0001 (one-

*Figure 5 continued on next page*

*Figure 5 continued*

way ANOVA with Dunnett's test). (**C**) Total RNA from RPE-1 cells transfected with the indicated siRNAs and treated with LLOMe for 2 hr was analyzed by RT-qPCR. Target mRNA levels were normalized to GAPDH mRNA levels; the expression levels in cells treated with control siRNA were set to 1. The individual values, mean, and SEM of relative mRNA levels are presented. The mean ± SEM values were calculated from three biological replicates. ****$p < 0.0001$ (one-way ANOVA with Dunnett's test). (**D, E**) HeLa cells were pre-treated with TAK-243 or HS-276 for 15 min, treated with LLOMe for 2 hr, and washed for 6 hr. Total cell lysates were subjected to immunoblotting with the indicated antibodies (**D**). Cells were stained with propidium iodide (PI), and the fractions of PI-positive cells were assessed using the flow cytometer. Error bars indicate SD (n=3). ****$p < 0.0001$ (one-way ANOVA with Dunnett's test) (**E**). (**F, G**) Total cell lysates from HeLa cells transfected with the indicated siRNAs, treated with LLOMe for 2 hr, and washed for 22 hr were subjected to immunoblotting with the indicated antibodies. (**H**) RPE-1 cells were stimulated for the indicated times with conditioned media (CM) from RPE-1 cells treated with LLOMe for 30 min and washed for 7.5 hr. Total cell lysates were subjected to immunoblotting with the indicated antibodies. (**I, J**) RPE-1 cells were stimulated for 15 min with CM from RPE-1 cells transfected with the indicated siRNAs, treated with LLOMe for 30 min, and washed for 7.5 hr. Total cell lysates were subjected to immunoblotting with the indicated antibodies.

The online version of this article includes the following source data and figure supplement(s) for figure 5:

**Source data 1.** Original files for western blot analysis for *Figure 5D, F, and G*.

**Source data 2.** Uncropped full images displayed in *Figure 5D, F, and G*.

**Source data 3.** Original files for western blot analysis for *Figure 5H–J*.

**Source data 4.** Uncropped full images displayed in *Figure 5H–J*.

**Figure supplement 1.** Anti-apoptotic regulation in the lysosomal damage response.

**Figure supplement 1—source data 1.** Original files for western blot analysis for *Figure 5—figure supplement 1A and B*.

**Figure supplement 1—source data 2.** Uncropped full images displayed in *Figure 5—figure supplement 1A and B*.

through the downstream effectors such as IKK and JNK in response to lysosomal damage, thereby promoting cell survival.

We further investigated the potential autocrine/paracrine-like effects of cytokines such as IL1β and IL6, which are expected to be secreted outside the cells due to their upregulation in the lysosomal damage response. IL6 activates the JAK–STAT pathway, which is accompanied by the phosphorylation of STAT3 (*Zhong et al., 1994*). Stimulation of untreated cells with conditioned media (CM) from lysosome-damaged cells resulted in the phosphorylation of STAT3, which indicates kinase activation (*Figure 5H*). These results suggest that the lysosomal damage response activates intercellular signaling involving the JAK–STAT pathway. Furthermore, the capacity to activate this intercellular signaling was decreased by depletion of TAB, TAK1, and IKKs (*Figure 5I and J*).

Collectively, these findings suggest that the K63 Ub–TAB–TAK1 pathway plays a pivotal role in cell survival and intercellular signaling in response to lysosomal damage.

## Discussion

In this study, comprehensive analyses of the transcriptome and proteome demonstrated that lysosomal damage causes substantial alterations in gene expression, leading to proteome remodeling. Furthermore, we identified the signal transduction pathway mediating this response (*Figure 6*). The accumulation of K63 ubiquitin chains in damaged lysosomes activates the TAB–TAK1–IKK–NF-κB pathway, which induces the expression of various transcription factors and cytokines. The present findings suggest that the ubiquitin-dependent signaling pathway plays a regulatory role in cell survival and intercellular signaling in the lysosomal damage response.

Prior to this study, our understanding of the lysosomal damage response was largely limited to the processes of repair and elimination of damaged lysosomes, as well as the activation and function of TFEB in lysosomal regeneration. Recent findings highlighting the role of SG formation in the lysosomal damage response suggest that additional pathways are involved in this process beyond what was previously anticipated (*Jia et al., 2022*). Nevertheless, the complete picture remains to be elucidated. This study provides new information on the regulation of gene expression and signal transduction in the lysosomal damage response that offers a novel perspective for future research.

Aberrations in the lysosomal damage response are relevant to the pathogenesis of various age-related diseases, including neurodegenerative diseases (*Tan and Finkel, 2023*; *Kakuda et al., 2024*). In the context of pathology, inflammation, which is associated with numerous diseases, is linked to lysosomes. The inflammatory response is not only regulated by lysosomal function but also triggered

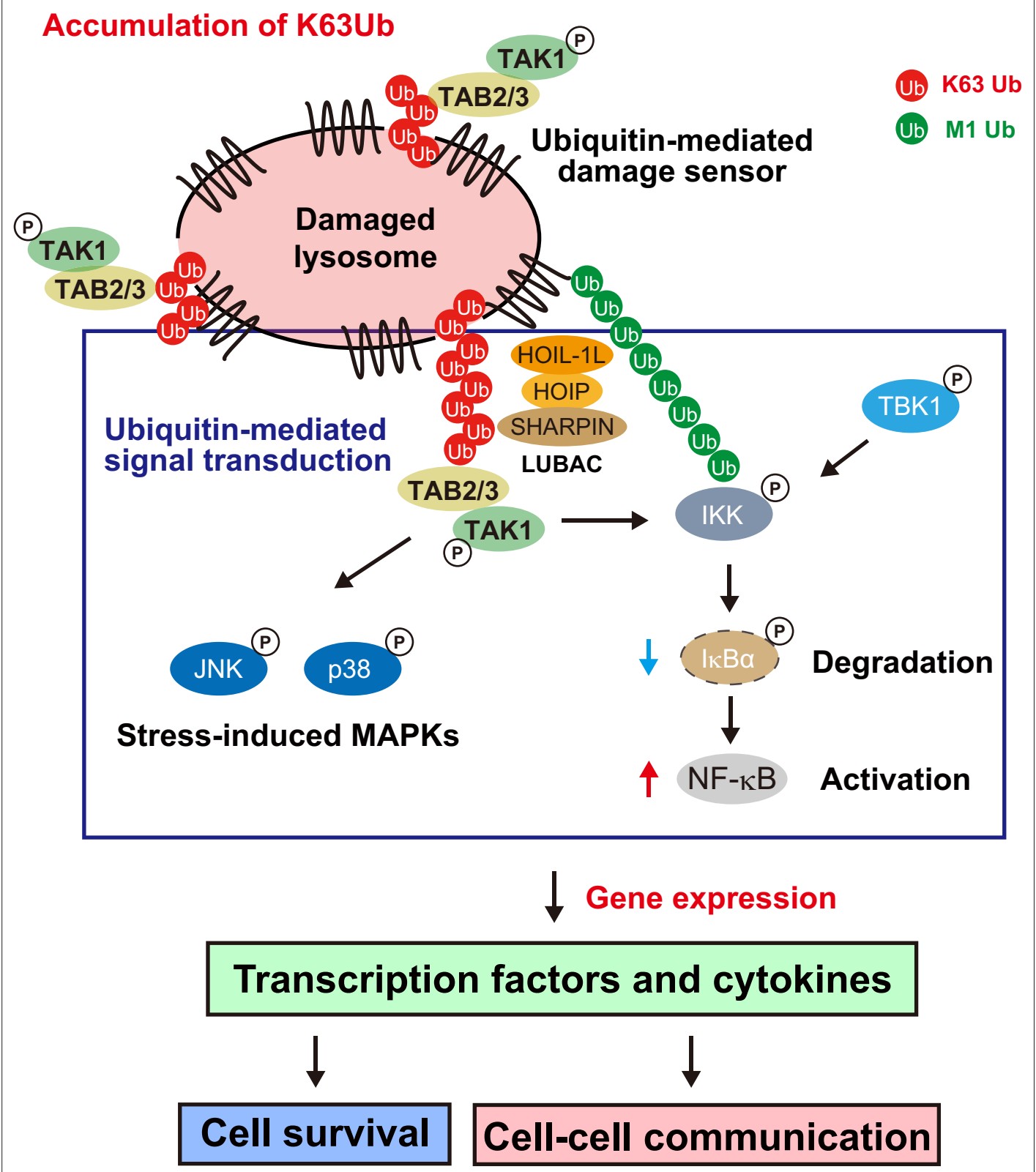

**Figure 6.** Model of the ubiquitin-mediated cellular response to lysosomal damage.

by lysosomal damage and observed in lysosomal storage diseases (*Simonaro, 2016*; *Seoane et al., 2020*). The NLRP3 inflammasome induces an inflammatory response upon lysosomal damage by increasing cytokine processing at the protein level (*Seoane et al., 2020*; *Muñoz-Planillo et al., 2013*). The ubiquitin-regulated gene expression of pro-inflammatory cytokines such as IL1β and IL6 may be involved in the induction of inflammation in the lysosomal damage response, in addition to the regulation at the stage of precursor processing. The present data on the regulation of gene expression by ubiquitin-mediated signal transduction may help elucidate the molecular mechanisms underlying diseases associated with lysosome dysfunction.

In this study, we demonstrated a pivotal role of the ubiquitin system in regulating gene expression in the lysosomal damage response, which extends its function beyond that in lysophagy initiation. We found that activation of the TAB–TAK1–IKK–NF-κB pathway, potentially by the accumulation of K63 ubiquitin chains in damaged lysosomes, and the subsequent induction of gene expression are crucial for cell survival and cell–cell communication in cultured cells. Further analysis is required to determine the impact of this action on disease pathogenesis and the maintenance of cellular homeostasis in organisms.

While we mainly focused on the TAB–TAK1–IKK–NF-κB pathway in this study, it should be noted that TAK1 also activates AMPK, subsequently promoting lysophagy (*Jia et al., 2020a*). AMPK activation by TAK1 may also contribute to NF-κB activation or distinct gene regulation. We found that TBK1 is required for NF-κB activation. In considering signal mediators such as AMPK and TBK1, it is postulated that cell homeostasis is cooperatively controlled by a set of regulators in the lysosomal damage response. A recent study demonstrated that linear ubiquitin chains formed by LUBAC and deubiquitinated by OTULIN activate NF-κB in the lysosomal damage response (*Zein et al., 2025*). We have confirmed that LUBAC is essential for NF-κB activation under our experimental conditions. It is well documented that LUBAC-catalyzed M1 ubiquitin chains recruit IKK subunits and transduce the signaling to downstream mediators in the canonical pathway. Therefore, we hypothesized that K63 ubiquitin chains in damage lysosomes initially activate the TAB–TAK1 pathway and trigger LUBAC-mediated M1 ubiquitination. We further postulate that M1 ubiquitination subsequently recruits the IKK complex to damaged lysosomes. Consequently, activated TAK1 phosphorylates IKK subunits, leading to NF-κB activation (*Figure 6*). Apart from the lysosomal damage response, activated STING at the Golgi also leads to NF-κB activation through LUBAC-mediated formation of linear ubiquitin chains (*Fischer et al., 2025*). In addition, the IKK complex is recruited to damaged mitochondria, potentially through its association with accumulated ubiquitin chains, and activates NF-κB (*Harding et al., 2023*). Although the involvement of the TAB–TAK1 pathway was not examined in these studies, a common mechanism underlying NF-κB activation induced by the accumulation of ubiquitin on damaged or stressed organelles may serve as a unifying determinant. Taken together with our previous findings that the accumulation of K63 ubiquitin chains in dysfunctional endosomes induces cytokine expression via the TAB–TAK1 pathway (*Endo et al., 2024a*), the present results suggest the existence of a universal response to the intracellular accumulation of K63 ubiquitin chains resulting from fluctuations in the cellular environment, including organelle stress. This universal response would entail the activation of NF-κB and the modulation of downstream target gene expression for the regulation of cellular functions that maintain homeostasis.

## Materials and methods

### Cell culture

RPE-1 (CRL-4000) and HeLa (CCL-2) cells, authenticated by STR profiling, were obtained from the American Type Culture Collection (ATCC). Low-passage cells were used in this study. All cells used for the experiment were mycoplasma-negative by the PCR test. RPE-1 cells were cultured at 37°C under 5% $CO_2$ in Dulbecco's Modified Eagle Medium (DMEM)/Ham's F-12 (WAKO, 048-29785) supplemented with 10% fetal bovine serum (FBS, Nichirei, 175013), 1 mM sodium pyruvate (Thermo Fisher Scientific, 11360070), 1× nonessential amino acids (Thermo Fisher Scientific, 11140050), and 1× Penicillin–Streptomycin–Glutamine (Thermo Fisher Scientific, 10378016). HeLa cells were cultured at 37°C under 5% $CO_2$ in DMEM (Thermo Fisher Scientific, D5796) supplemented with 10% FBS, 1 mM sodium pyruvate, 1× nonessential amino acids, and 1× Penicillin–Streptomycin–Glutamine. For experiments using conditioned medium, RPE-1 cells were stimulated with culture medium from RPE-1 cells that

had been treated with or without 1 mM LLOMe (Cayman, 16008). At 30 min after treatment, the LLOMe was washed out and cells were further incubated in DMEM supplemented with 0.5% FBS, 1 mM sodium pyruvate, 1× nonessential amino acids, and 1× Penicillin–Streptomycin–Glutamine for 8 hr. HeLa cells stably expressing siRNA-resistant FLAG-TAB2 WT, dNZF, and E685A were generated previously (*Endo et al., 2024a*).

## Reagents and inhibitors

To induce lysosomal damage, cells were treated with 1 mM LLOMe, 200 μM GPN (Cayman, 14634), and DC661 (Cayman, 34899) for the indicated times. In the event of a longer incubation period, the LLOMe was washed out 2 hr after treatment. The following inhibitors were used in this study: TAK-243 (10 μM, Active Biochem, A-1384), HS-276 (10 μM, Sigma-Aldrich, SML-3629), BX-795 (2 μM, Abcam, ab142016), and JNK-IN-8 (10 μM, Cayman, 18096).

## Transcriptome analysis

Total RNA from RPE-1 cells was prepared using the Qiashredder kit (QIAGEN, 79654), the RNeasy kit (QIAGEN, 74104), and DNase (QIAGEN, 79254). The quality of RNA samples was assessed using an Agilent 2100 Bioanalyzer (Agilent Technologies); an RNA integrity number >9 was recorded for all samples. A total of 500 ng of each sample was processed with the Illumina Stranded mRNA Prep kit, and indexes were added with the IDT for Illumina RNA UD Indexes Set A (Illumina). The average library size was estimated using the Agilent 2100 Bioanalyzer, and libraries were quantified using a Qubit Fluorometer (Thermo Fisher Scientific). Samples were pooled into a single library and sequenced using the NextSeq 1000/2000 P2 Reagents (300 cycles, paired-end 150 bp) on the Illumina NextSeq 1000 system. Sequence data were analyzed using the onboard DRAGEN RNA Pipeline Application (version 3.10.12).

## Proteome analysis

RPE-1 cells were lysed in 5% SDS lysis buffer (5% SDS and 50 mM triethylammonium bicarbonate [TEAB, Thermo Fisher Scientific, 90114]), followed by sonication with an ultrasonication probe. Cell lysates were reduced and alkylated with 4.5 mM dithiothreitol (Thermo Fisher Scientific, A39255) for 30 min at 55°C and 10 mM iodoacetamide (Thermo Fisher Scientific, A39271) for 15 min at room temperature (RT). Subsequently, 30 μg of cell lysates were loaded onto S-Trap micro columns (ProtiFi, C02-micro); trypsin and lysyl-endopeptidase (Lys-C) solution (1:10 wt/wt, Thermo Fisher Scientific, A41009) was added and incubated for 16 hr at 37°C. The eluted peptides were dried in a vacuum concentrator and resuspended in 0.1% trifluoroacetic acid (TFA). Aliquots containing 500 ng of peptides from each sample were loaded onto a Vanquish Neo UHPLC system-connected Orbitrap Exploris 480 mass spectrometer (Thermo Fisher Scientific). The peptides were separated on an analytical column (C18, 1.7 μm particle size × 75 μm diameter × 600 mm, IonOpticks, AUR3-60075C18) heated at 55°C in a column oven (Sonation, PRSO-C2) with a constant flow rate of 250 nL/min. The peptides were eluted with a 0–40% acetonitrile gradient over 120 min. Peptide ionization was performed using the Nanospray Flex Ion Source (Thermo Fisher Scientific). The Orbitrap Exploris 480 mass spectrometer was operated in data-independent acquisition (DIA) mode using a full scan (m/z range of 380–985, nominal resolution of 60,000, and target value of $3 \times 10^6$ ions) followed by DIA-mass spectrometry (MS) scans (fixed collision energy of 30%, isolation width of 10 m/z with an overlapping of 1 m/z, nominal resolution of 15,000, and target value of $2 \times 10^6$ ions).

## DIA-MS data processing and visualization

The MS raw files were searched against the human UniProt reference proteome (Uniprot ID: UP000005640, reviewed, canonical, 20,563 entries) in library-free mode using DIA-NN software (version 1.9.1) (*Demichev et al., 2020*). The parameters that differed from the default settings were as follows: variable modifications, oxidation of methionine, and acetylation of the peptide N-terminus; precursor m/z range, 380–985. Following processing with DIA-NN, the exported data were subjected to imputation using Perseus software (*Tyanova et al., 2016*). The default setting was used, assuming that missing values represent low protein abundance. In this case, missing values were replaced by random numbers drawn from a normal distribution across the entire matrix. Gene enrichment tests and clustering were performed using the RNAseqChef software (*Etoh and Nakao, 2023*), and heatmaps

were generated using the Morpheus platform (https://software.broadinstitute.org/morpheus/). The volcano plots, scatter plots, and bar graphs were visualized using GraphPad Prism software (version 8.1.0, GraphPad).

## Cell lysis and immunoblotting

Cells were lysed with 2% SDS lysis buffer (2% SDS, 20 mM HEPES pH 7.5, 1 mM EDTA) containing a complete protease inhibitor cocktail (EDTA-free, Roche, 05056489001) and a phosphatase inhibitor cocktail (Roche, 4906845001). Cell lysates were boiled in 1× LDS NuPAGE sample buffer (Thermo Fisher Scientific, NP0008) for 10 min at 70°C, and then electrophoresed on 4–12% NuPAGE Bis-Tris gels (Thermo Fisher Scientific). Proteins were transferred to polyvinylidene difluoride membranes (Millipore, IPVH00010 or Pall, EH-2222). The membranes were incubated with 5% nonfat milk for 1 hr at RT, followed by incubation for 2 hr at RT with primary antibodies. The primary antibodies used for immunoblotting were anti-IL1β mouse monoclonal (Cell Signaling Technology, 12242), anti-IL6 rabbit monoclonal (Cell Signaling Technology, 12153), anti-IRF1 rabbit monoclonal (Cell Signaling Technology, 8478), anti-NKX3.1 rabbit monoclonal (Cell Signaling Technology, 92998), anti-c-Fos rabbit monoclonal (Cell Signaling Technology, 2250), anti-c-Jun rabbit monoclonal (Cell Signaling Technology, 9165), HRP-conjugated anti-α-tubulin rabbit polyclonal (MBL, PM054-7), anti-K63 ubiquitin rabbit monoclonal (Millipore, 05-1308), anti-ubiquitin mouse monoclonal (Santa Cruz Biotechnology, sc-8017), anti-phospho-TAK1 (T184/T187) rabbit monoclonal (Cell Signaling Technology, 4508), anti-TAK1 rabbit polyclonal (Cell Signaling Technology, 4505), anti-TAB2 rabbit polyclonal (Abcam, ab222214), anti-TAB3 rabbit polyclonal (Abcam, ab85655), HRP-conjugated anti-FLAG mouse monoclonal (Sigma-Aldrich, A8592), anti-TFEB rabbit monoclonal (Cell Signaling Technology, 37785), anti-phospho-IKKα/β (S176/S180) rabbit monoclonal (Cell Signaling Technology, 2697), anti-IKKα rabbit polyclonal (Cell Signaling Technology, 2682), anti-IKKβ rabbit monoclonal (Cell Signaling Technology, 8943), anti-phospho-IκBα (S32) rabbit monoclonal (Cell Signaling Technology, 2859), anti-IκBα rabbit monoclonal (Cell Signaling Technology, 4812), anti-phospho-JNK (T183/Y185) rabbit monoclonal (Cell Signaling Technology, 4668), anti-JNK rabbit polyclonal (Cell Signaling Technology, 9252), anti-phospho-p38 (T180/Y182) rabbit polyclonal (Cell Signaling Technology, 9211), anti-p38 rabbit monoclonal (Cell Signaling Technology, 8690), anti-phospho-ERK (T202/Y204) rabbit monoclonal (Cell Signaling Technology, 4370), anti-ERK rabbit monoclonal (Cell Signaling Technology, 4695), anti-cleaved caspase-3 rabbit polyclonal (Cell Signaling Technology, 9661), anti-phospho-STAT3 (Y705) rabbit monoclonal (Cell Signaling Technology, 9145), anti-STAT3 rabbit monoclonal (Cell Signaling Technology, 4904), anti-phospho-TBK1 (S172) rabbit monoclonal (Cell Signaling Technology, 5483), anti-TBK1 rabbit monoclonal (Abcam, ab40676), and anti-RNF31 rabbit monoclonal (Cell Signaling Technology, 99633) antibodies. The membranes were then incubated with secondary antibodies for 50 min at RT. The secondary antibodies used were HRP-conjugated goat anti-rabbit Ig (Promega, W4011) and anti-mouse Ig (Promega, W4021). The antibodies were diluted with Can Get Signal Solutions (TOYOBO, NKB-101). Chemiluminescence images were developed using the ECL Prime Western Blotting Detection Reagent (Cytiva, RPN2236) and acquired with a Fusion FX7 (Vilber Bio Imaging).

## Phosphoproteome analysis

RPE-1 cells were treated with or without 1 mM LLOMe and 10 µM HS-276 for the indicated times, after which they were lysed, reduced, and alkylated as described above (proteome analysis). Aliquots containing 200 µg of lysate were loaded onto S-Trap mini columns (ProtiFi, C02-mini), followed by the addition of the trypsin solution (1:10 wt/wt, Cell Signaling Technology, 56296) and incubation for 16 hr at 37°C. The eluted peptides were dried in a vacuum concentrator. Phosphopeptides were enriched using the High-Select Fe-NTA Phosphopeptide Enrichment Kit (Thermo Fisher Scientific, A32992) and dried in a vacuum concentrator. The resultant peptides were cleaned using C18 spin tips (Thermo Fisher Scientific, 84850), dried in a vacuum concentrator, resuspended in 0.1% TFA, and loaded onto a Vanquish Neo UHPLC system-connected Orbitrap Exploris 480 mass spectrometer. The peptides were separated on an analytical column (C18, 1.7 µm particle size × 75 µm diameter × 250 mm, IonOpticks, AUR3-25075C18) heated at 55°C in a column oven with a constant flow rate of 250 nL/min. The peptides were eluted with a 0–40% acetonitrile gradient over 120 min. Peptide ionization was performed using the Nanospray Flex

Ion Source. The Orbitrap Exploris 480 mass spectrometer was operated in data-dependent acquisition (DDA) mode utilizing a full scan (m/z range, 375–1500; nominal resolution, 60,000; target value, $3 \times 10^6$ ions) followed by MS/MS scans (fixed collision energy, 30%; isolation width, 1.6 m/z; nominal resolution, 15,000, and target value, $1 \times 10^5$ ions). Precursor ions selected for fragmentation (charge state 2–6) were placed on a dynamic exclusion list for 20 s. A '1 s cycle' DDA method was used, whereby the most intense ions were selected every second for MS/MS fragmentation by higher-energy collisional dissociation.

## DDA-MS data processing and visualization

The MS raw files were searched against the human UniProt reference proteome (Uniprot ID: UP000005640, reviewed, canonical, 20,563 entries) using the Sequest HT search program in Proteome Discoverer 3.1 (Thermo Fisher Scientific). Intensity-based non-label quantification was performed using the Precursor Ions Quantifier node in Proteome Discoverer 3.1. Volcano plots, scatter plots, and bar graphs were visualized using GraphPad Prism (version 8.1.0, GraphPad).

## Immunofluorescence and confocal microscopy analysis

Cells were plated in 35 mm glass-bottomed dishes coated with poly-L-lysine (MatTek, P35-GC-0-10-C). They were fixed with 4% paraformaldehyde in PBS for 10 min at RT and permeabilized with 0.2% Triton X-100 in PBS for 5 min at RT. Alternatively, the cells were incubated with ice-cold methanol for 10 min on ice. Following incubation with 5% FBS and 0.1% Tween in PBS for 30 min at RT, cells were incubated with primary antibodies for 2 hr at RT and then stained with secondary antibodies and DAPI (Thermo Fisher Scientific, D1306) for 1 hr at RT. The following primary antibodies were used: anti-TAB2 mouse monoclonal (Santa Cruz Biotechnology, sc-398188), anti-LAMP1 rabbit monoclonal (Cell Signaling Technology, 9091), anti-K63 ubiquitin rabbit monoclonal (Millipore, 05-1308), anti-TAK1 mouse monoclonal (Santa Cruz Biotechnology, sc-7967), and anti-Galectin-3 rat monoclonal (Santa Cruz Biotechnology, sc-23938). The following secondary antibodies were purchased from Thermo Fisher Scientific: Alexa Fluor 488-conjugated anti-mouse (A-11029) and anti-rat (A-11006), and Alexa Fluor 594-conjugated anti-rabbit (A-11012). After staining, the cells were coverslipped (Matsunami, C015001) with SlowFade Gold (Thermo Fisher Scientific, S36936). Images were captured using ZEN 3.8 imaging software and LSM980 laser-scanning confocal microscopes equipped with a Plan-Apochromat 63×/1.4NA oil lens (Carl Zeiss).

## siRNA transfection

Cells were transfected with siRNAs using Lipofectamine RNAiMax (Thermo Fisher Scientific) at a final siRNA concentration of 30 nM. Cells were harvested and analyzed 72 hr after transfection with siRNA. The siRNAs utilized pools of four different sequences (Horizon Discovery). The following siRNAs were used: TAB2 siRNAs: L-004771; TAB3 siRNAs: L-015572; TAK1 siRNAs: L-003790; IKKα siRNAs: L-003473; IKKβ siRNAs: L-003503; IKKγ siRNAs: L-003763; and RNF31 siRNAs: L-021419.

## RT-qPCR analysis

Total RNA was extracted using NucleoSpin RNA Plus (MACHEREY-NAGEL, 740984.250). cDNA was generated using the ReverTra Ace qPCR RT Master Mix with gDNA Remover kit (TOYOBO, FSQ-301). Quantification of mRNA was performed using a Light Cycler 480 (Roche) with THUNDERBIRD SYBR qPCR Mix (TOYOBO, QPS-201) as the detection reagent. The primer sets used for qRT-PCR were as follows: IL1β forward: TACGATCACTGAACTGCACGC, IL1β reverse: CTTGTTGCTCCATATCCTGTCCC; IL6 forward: TCATCACTGGTCTTTTGGAGTTTG, IL6 reverse: CAGCTCTGGCTTGTTCCTCAC; IRF1 forward: GCCATTCACACAGGCCGATAC, IRF1 reverse: TGCTCTGGTCTTTCACCTCCTC; NKX3.1 forward: CTGGGAGACTTGGAGAAGCAC, NKX3.1 reverse: GGATAGCTGTTATACACGGAGACC; and GAPDH forward: AGAAGGTGGTGAAGCAGGCG, GAPDH reverse: CAAAGTGGTCGTTGAGGGCAATG.

## Cell death assay

Cell death upon lysosomal damage was assessed by propidium iodide (PI) staining. Both cells attached to the culture dish and suspended in media were collected and stained with 50 µg/mL of PI (Thermo Fisher Scientific, P3566) for 15 min at RT. After washing excess PI, cells were resuspended in fresh PBS. Stained cells were loaded and counted on LSR-Fortessa (BD Biosciences).

## Statistical analysis

Statistical analysis was performed using GraphPad Prism (version 8.1.0). All statistical information is provided in the figure legends. First, the sample distribution was assessed using the Shapiro–Wilk test. In the absence of formal testing, datasets with small sample sizes were assumed to have a normal distribution. The unpaired two-tailed Student's t-test was used to determine statistical significance when comparing unpaired two independent groups with normal distribution and no significant difference in standard deviation (SD). For multiple comparisons involving more than two unpaired groups with normal distribution, an ordinary one-way ANOVA with Dunnett's multiple comparison test was used. In all instances, statistical significance was evaluated with a 95% confidence interval, and a p-value$<0.05$ was considered statistically significant.

## Acknowledgements

This research was supported by JSPS KAKENHI (grant no. JP18K14623 and JP20K06568 to AE, JP23H04923 and JP23K23841 to KY, and JP23H04921 to KT) and AMED (grant no. JP21gm6410012 to AE).

## Additional information

### Funding

| Funder | Grant reference number | Author |
|---|---|---|
| Japan Society for the Promotion of Science | JP18K14623 | Akinori Endo |
| Japan Society for the Promotion of Science | JP20K06568 | Akinori Endo |
| Japan Society for the Promotion of Science | JP23H04923 | Koji Yamano |
| Japan Society for the Promotion of Science | JP23K23841 | Koji Yamano |
| Japan Society for the Promotion of Science | JP23H04921 | Keiji Tanaka |
| Japan Agency for Medical Research and Development | JP21gm6410012 | Akinori Endo |
| Japan Agency for Medical Research and Development | JP24gm14100003 | Yukiko Yoshida |
| Japan Society for the Promotion of Science | JP22H02305 | Yukiko Yoshida |

The funders had no role in study design, data collection and interpretation, or the decision to submit the work for publication.

### Author contributions

Akinori Endo, Conceptualization, Supervision, Funding acquisition, Investigation, Visualization, Writing – original draft, Project administration, Writing – review and editing; Chikage Takahashi, Naoko Ishibashi, Yasumasa Nishito, Investigation; Koji Yamano, Funding acquisition, Investigation; Keiji Tanaka, Supervision, Funding acquisition; Yukiko Yoshida, Supervision, Project administration, Writing – review and editing

### Author ORCIDs

Akinori Endo https://orcid.org/0000-0003-0225-4832
Koji Yamano https://orcid.org/0000-0002-4692-161X
Yukiko Yoshida https://orcid.org/0000-0002-0629-0219

Reviewer #1 (Public review): https://doi.org/10.7554/eLife.106901.3.sa1
Reviewer #2 (Public review): https://doi.org/10.7554/eLife.106901.3.sa2
Reviewer #3 (Public review): https://doi.org/10.7554/eLife.106901.3.sa3
Author response https://doi.org/10.7554/eLife.106901.3.sa4

# Additional files

## Supplementary files

Supplementary file 1. Transcriptome analysis L-leucyl–L-leucine methyl ester (LLOMe) 2 hr.

Supplementary file 2. Proteome analysis L-leucyl–L-leucine methyl ester (LLOMe) 2 hr.

Supplementary file 3. Correlation of transcriptome and proteome L-leucyl–L-leucine methyl ester (LLOMe) 2 hr.

Supplementary file 4. Transcriptome analysis RNAi L-leucyl–L-leucine methyl ester (LLOMe) 2 hr.

Supplementary file 5. Proteome analysis L-leucyl–L-leucine methyl ester (LLOMe) 30 min.

Supplementary file 6. Phosphoproteome analysis with HS-276.

MDAR checklist

## Data availability

The RNA sequencing data have been deposited with links to BioProject accession numbers PRJDB19809 and PRJDB19818 and in the BioProject database of DNA Data Bank of Japan (DDBJ). The mass spectrometry proteomics data have been deposited to the ProteomeXchange Consortium via the PRIDE partner repository (*Perez-Riverol et al., 2022*) with dataset identifiers PXD058072 (related to *Figure 1B–E* and *Figure 1—figure supplement 1A and B*), PXD058075 (related to *Figure 4A and B*), and PXD058080 (related to *Figure 4—figure supplement 1A–C*). Source data files containing original images and uncropped full images have been provided for all the western blot analyses.

The following datasets were generated:

| Author(s) | Year | Dataset title | Dataset URL | Database and Identifier |
|---|---|---|---|---|
| Endo A | 2025 | Proteome changes of RPE-1 cells in response to lysosomal damage | https://www.ebi.ac.uk/pride/archive/projects/PXD058072 | PRIDE, PXD058072 |
| Endo A | 2025 | Proteome changes of RPE-1 cells in response to lysosomal damage with the TAK1 inhibitor | https://www.ebi.ac.uk/pride/archive/projects/PXD058075 | PRIDE, PXD058075 |
| Endo A | 2025 | Phosphoproteome changes of RPE-1 cells in response to lysosomal damage with the TAK1 inhibitor | https://www.ebi.ac.uk/pride/archive/projects/PXD058080 | PRIDE, PXD058080 |
| Endo A, Nishito Y | 2025 | RNA sequencing, transcriptomic alternations in response to lysosomal damage | https://ddbj.nig.ac.jp/search/entry/bioproject/PRJDB19809 | DDBJ, PRJDB19809 |
| Endo A, Nishito Y | 2025 | RNA sequencing, TAB- and TAK1-dependent transcriptomic alternations in response to lysosomal damage | https://ddbj.nig.ac.jp/search/entry/bioproject/PRJDB19818 | DDBJ, PRJDB19818 |

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

# Appendix 1

## Appendix 1—key resources table

| Reagent type (species) or resource | Designation | Source or reference | Identifiers | Additional information |
|---|---|---|---|---|
| Cell line (*Homo sapiens*) | RPE-1 | ATCC | CRL-4000 RRID:CVCL_4388 | |
| Cell line (*Homo sapiens*) | HeLa | ATCC | CCL-2 RRID:CVCL_0030 | |
| Antibody | anti-IL6 (rabbit monoclonal) | Cell Signaling Technology | Cat#: 12153 RRID:AB_2687897 | WB (1:1000) |
| Antibody | anti-IRF1 (rabbit monoclonal) | Cell Signaling Technology | Cat#: 8478 RRID:AB_10949108 | WB (1:1000) |
| Antibody | anti-NKX3.1 (rabbit monoclonal) | Cell Signaling Technology | Cat#: 92998; RRID:AB_2800197 | WB (1:1000) |
| Antibody | anti-c-Fos (rabbit monoclonal) | Cell Signaling Technology | Cat#: 2250 RRID:AB_2247211 | WB (1:1000) |
| Antibody | anti-c-Jun (rabbit monoclonal) | Cell Signaling Technology | Cat#: 9165 RRID:AB_2130165 | WB (1:1000) |
| Antibody | HRP-conjugated anti-α-tubulin (rabbit polyclonal) | MBL | Cat#: PM054-7 RRID:AB_10695326 | WB (1:1000) |
| Antibody | anti-K63 ubiquitin (rabbit monoclonal) | Millipore | Cat#: 05-1308 RRID:AB_1587580 | WB (1:500) IF (1:200) |
| Antibody | anti-ubiquitin (mouse monoclonal) | Santa Cruz Biotechnology | Cat# sc-8017 RRID:AB_628423 | WB (1:200) |
| Antibody | anti-phospho-TAK1 (T184/T187) (rabbit monoclonal) | Cell Signaling Technology | Cat#: 4508 RRID:AB_561317 | WB (1:1000) |
| Antibody | anti-TAK1 (rabbit polyclonal) | Cell Signaling Technology | Cat#: 4505 RRID:AB_490858 | WB (1:1000) |
| Antibody | anti-TAB2 (rabbit polyclonal) | Abcam | Cat#: ab222214 | WB (1:500) |
| Antibody | anti-TAB3 (rabbit polyclonal) | Abcam | Cat#: ab85655 RRID:AB_2140510 | WB (1:500) |
| Antibody | HRP-conjugated anti-FLAG (mouse monoclonal) | Sigma-Aldrich | Cat#: A8592 RRID:AB_439702 | WB (1:1000) |
| Antibody | anti-TFEB (rabbit monoclonal) | Cell Signaling Technology | Cat#: 37785 RRID:AB_2799119 | WB (1:1000) |
| Antibody | anti-phospho-IKKα/β (S176/S180) (rabbit monoclonal) | Cell Signaling Technology | Cat#: 2697 RRID:AB_2079382 | WB (1:1000) |
| Antibody | anti-IKKα (rabbit polyclonal) | Cell Signaling Technology | Cat#: 2682 RRID:AB_331626 | WB (1:1000) |
| Antibody | anti-IKKβ (rabbit monoclonal) | Cell Signaling Technology | Cat#: 8943 RRID:AB_11024092 | WB (1:1000) |
| Antibody | anti-phospho-IκBα (S32) (rabbit monoclonal) | Cell Signaling Technology | Cat#: 2859 RRID:AB_561111 | WB (1:1000) |
| Antibody | anti-IκBα (rabbit monoclonal) | Cell Signaling Technology | Cat#: 4812 RRID:AB_10694416 | WB (1:1000) |
| Antibody | anti-phospho-JNK (T183/Y185) (rabbit monoclonal) | Cell Signaling Technology | Cat#: 4668 RRID:AB_823588 | WB (1:1000) |
| Antibody | anti-JNK (rabbit polyclonal) | Cell Signaling Technology | Cat#: 9252 RRID:AB_2250373 | WB (1:1000) |
| Antibody | anti-phospho-p38 (T180/Y182) (rabbit polyclonal) | Cell Signaling Technology | Cat#: 9211 RRID:AB_331641 | WB (1:1000) |
| Antibody | anti-p38 (rabbit monoclonal) | Cell Signaling Technology | Cat#: 8690 RRID:AB_10999090 | WB (1:1000) |
| Antibody | anti-phospho-ERK (T202/Y204) (rabbit monoclonal) | Cell Signaling Technology | Cat#: 4370 RRID:AB_2315112 | WB (1:1000) |

*Appendix 1 Continued on next page*

*Appendix 1 Continued*

| Reagent type (species) or resource | Designation | Source or reference | Identifiers | Additional information |
|---|---|---|---|---|
| Antibody | anti-ERK (rabbit monoclonal) | Cell Signaling Technology | Cat#: 4695 RRID:AB_390779 | WB (1:1000) |
| Antibody | anti-cleaved caspase-3 (rabbit polyclonal) | Cell Signaling Technology | Cat#: 9661 RRID:AB_2341188 | WB (1:1000) |
| Antibody | anti-phospho-STAT3 (Y705) (rabbit monoclonal) | Cell Signaling Technology | Cat#: 9145 RRID:AB_2491009 | WB (1:1000) |
| Antibody | anti-STAT3 (rabbit monoclonal) | Cell Signaling Technology | Cat#: 4904 RRID:AB_331269 | WB (1:1000) |
| Antibody | anti-phospho-TBK1 (S172) (rabbit monoclonal) | Cell Signaling Technology | Cat#: 5483 RRID:AB_10693472 | WB (1:1000) |
| Antibody | anti-TBK1 (rabbit monoclonal) | Abcam | Cat#: ab40676 RRID:AB_776632 | WB (1:2000) |
| Antibody | anti-RNF31 (rabbit monoclonal) | Cell Signaling Technology | Cat#: 99633 RRID:AB_2891320 | WB (1:1000) |
| Antibody | HRP-conjugated goat anti-rabbit IgG | Promega | Cat#: W4011 RRID:AB_430833 | WB (1:20,000) |
| Antibody | HRP-conjugated goat anti-mouse IgG | Promega | Cat#: W4021 RRID:AB_430834 | WB (1:20,000) |
| Antibody | anti-TAB2 (mouse monoclonal) | Santa Cruz Biotechnology | Cat#: sc-398188 RRID:AB_2885043 | IF (1:50) |
| Antibody | anti-LAMP1 (rabbit monoclonal) | Cell Signaling Technology | Cat#: 9091 RRID:AB_2687579 | IF (1:100) |
| Antibody | anti-TAK1 (mouse monoclonal) | Santa Cruz Biotechnology | Cat#: sc-7967 RRID:AB_627929 | IF (1:50) |
| Antibody | anti-Galectin-3 (rat monoclonal) | Santa Cruz Biotechnology | Cat#: sc-23938 RRID:AB_627658 | WB (1:50) |
| Antibody | Alexa Fluor 488-conjugated anti-mouse (goat polyclonal) | Thermo Fisher Scientific | Cat#: A-11029 RRID:AB_2534088 | IF (1:1000) |
| Antibody | Alexa Fluor 488-conjugated anti-rat (goat polyclonal) | Thermo Fisher Scientific | Cat#: A-11006 RRID:AB_2534074 | IF (1:1000) |
| Antibody | Alexa Fluor 594-conjugated anti-rabbit (goat polyclonal) | Thermo Fisher Scientific | Cat#: A-11012 RRID:AB_2534079 | IF (1:1000) |
| Sequence-based reagent | siRNA: non-targeting | Horizon Discovery | Cat#: D-001810-03 | On-TARGETplus |
| Sequence-based reagent | siRNA: TAB2 | Horizon Discovery | Cat#: L-004771 | On-TARGETplus SMARTpool |
| Sequence-based reagent | siRNA: TAB3 | Horizon Discovery | Cat#: L-015572 | On-TARGETplus SMARTpool |
| Sequence-based reagent | siRNA: TAK1 | Horizon Discovery | Cat#: L-003790 | On-TARGETplus SMARTpool |
| Sequence-based reagent | siRNA: IKKα | Horizon Discovery | Cat#: L-003473 | On-TARGETplus SMARTpool |
| Sequence-based reagent | siRNA: IKKβ | Horizon Discovery | Cat#: L-003503 | On-TARGETplus SMARTpool |
| Sequence-based reagent | siRNA: IKKγ | Horizon Discovery | Cat#: L-003763 | On-TARGETplus SMARTpool |
| Sequence-based reagent | siRNA: RNF31 | Horizon Discovery | Cat#: L-021419 | On-TARGETplus SMARTpool |
| Commercial assay or kit | RNeasy kit | QIAGEN | Cat#: 74104 | |
| Commercial assay or kit | DNase | QIAGEN | Cat#: 79254 | |
| Commercial assay or kit | S-Trap micro columns | ProtiFi | Cat#: C02-micro | |
| Commercial assay or kit | S-Trap mini columns | ProtiFi | Cat#: C02-mini | |
| Commercial assay or kit | S-Trap mini columns | ProtiFi | Cat#: C02-mini | |

*Appendix 1 Continued on next page*

*Appendix 1 Continued*

| Reagent type (species) or resource | Designation | Source or reference | Identifiers | Additional information |
|---|---|---|---|---|
| Commercial assay or kit | Trypsin and lysyl-endopeptidase (Lys-C) | Thermo Fisher Scientific | Cat#: A41009 | |
| Commercial assay or kit | High-Select Fe-NTA Phosphopeptide Enrichment Kit | Thermo Fisher Scientific | Cat#: A32992 | |
| Commercial assay or kit | C18 spin tips | Thermo Fisher Scientific | Cat#: 84850 | |
| Chemical compound, drug | L-Leucyl–L-Leucine methyl ester (LLOMe) | Cayman | Cat#: 16008 | |
| Chemical compound, drug | Glycyl-L-phenylalanine 2-naphthylamide (GPN) | Cayman | Cat#: 14634 | |
| Chemical compound, drug | TAK-243 | Active Biochem | Cat#: A-1384 | |
| Chemical compound, drug | HS-276 | Sigma-Aldrich | Cat#: SML-3629 | |
| Chemical compound, drug | BX-795 | Abcam | Cat#: ab142016 | |
| Chemical compound, drug | JNK-IN-8 | Cayman | Cat#: 18096 | |
| Software, algorithm | DRAGEN RNA Pipeline Application | Illumina | | v.3.10.12 |
| Software, algorithm | DIA-NN | *Demichev et al., 2020* | RRID:SCR_022865 | v.1.9.1 |
| Software, algorithm | Perseus | *Tyanova et al., 2016* | RRID:SCR_015753 | v.2.0.3 |
| Software, algorithm | RNAseqChef | *Etoh and Nakao, 2023* | | v.1.1.4 |
| Software, algorithm | Morpheus | https://software.broadinstitute.org/morpheus/ | RRID:SCR_017386 | |
| Software, algorithm | GraphPad Prism | *Demichev et al., 2020* | RRID:SCR_002798 | v.8.1.0 |
| Software, algorithm | Proteome Discoverer | Thermo Fisher Scientific | RRID:SCR_014477 | v.3.1 |
| Software, algorithm | ZEN | Carl Zeiss | RRID:SCR_013672 | v.3.8 |

