## [Editor Report · eLife Assessment]

This study presents the **important** finding that lysosomal damage triggers inflammatory signaling through ubiquitination and the TAB-TAK1-IKK-NF-kB axis. The data obtained from the unbiased transcriptomic and proteomic analyses are **convincing** and provide invaluable information to the field. Although further experiments will be required to clarify how TAB2/3 are recruited after various types of lysosome damage, this work will be of interest to researchers in the fields of organelle biology and inflammation.

---

## [Referee Report · Reviewer #1 (Public review)]

Summary:

Lysosomal damage is commonly found in many diseases including normal aging and age-related disease. However, the transcriptional programs activated by lysosomal damage has not been thoroughly characterized. This study aims to investigate lysosome damage-induced major transcriptional responses and the underlying signaling basis. The authors have convincingly shown that lysosomal damage activates a ubiquitination-dependent signaling axis involving TAB, TAK1, and IKK, which culminate in the activation of NF-kB and subsequent transcriptional upregulation of pro-inflammatory genes and pro-survival genes. Overall, the major aims of this study are successfully achieved.

Strengths:

This study is well-conceived and strictly executed, leading to clear and well-supported conclusions. Through unbiased transcriptomics and proteomics screens, the authors identifies NF-kB as a major transcriptional program activated upon lysosome damage. TAK1 activation by lysosome damage-induced ubiquitination is found to be essential for NF-kB activation and MAP kinase signaling. The transcriptional and proteomic changes are shown to be largely driven by TAK1 signaling. Finally, the TAK1-IKK signaling is shown to provide resistance to apoptosis during lysosomal damage response. The main signaling axis of this pathway has been convincingly demonstrated.

Overall, this study identifies major transcriptional responses following lysosomal damage through unbiased approaches. It is important to consider the impact of these pathways in disease settings where lysosomal integrity is compromised.

Comments on revisions:

The authors have adequately addressed all previous comments. I have no further recommendations.

---

## [Referee Report · Reviewer #2 (Public review)]

Summary:

Endo et al. investigate the novel role of ubiquitin response upon lysosomal damage in activating cellular signaling for cell survival. The authors provide a comprehensive transcriptome and proteome analysis of aging-related cells experiencing lysosomal damage, identifying transcription factors involved in transcriptome and proteome remodeling with a focus on the NF-κB signaling pathway. They further characterized the K63-ubiquitin-TAB-TAK1-NF-κB signaling axis in controlling gene expression, inflammatory responses, and apoptotic processes.

Strengths:

In the aging-related model, the authors provide a comprehensive transcriptome and characterize the K63-ubiquitin-TAB-TAK1-NF-κB signaling axis. Through compelling experiments and advanced tools, they elucidate its critical role in controlling gene expression, inflammatory responses, and apoptotic processes.

Weaknesses:

The study lacks deeper connections with previous research, particularly:

• The established role of TAB-TAK1 in AMPK activation during lysosomal damage

• The potential significance of TBK1 in NF-κB signaling pathways

Comments on revisions:

The authors have successfully addressed all the raised questions and the manuscript is now significantly improved.

---

## [Referee Report · Reviewer #3 (Public review)]

Summary:

The response to lysosomal damage is a fast-moving and timely field. Besides repair and degradation pathways, increasing interest has been focusing on damaged-induced signaling. The authors conducted both transcriptomics and proteomics to characterize the cellular response to lysosomal damage. They identify a signaling pathway leading to activation of NFkappaB. Based on this and supported by Western blot and microscopy data, the authors nicely show that TAB2/3 and TAK1 are activated at damaged lysosomes and kick off the pathway to alter gene expression, which induces cytokines and protect from cell death. TAB2/3 activation is proposed to occur through K63 ubiquitin chain formation. Generally, this is a careful and well conducted study that nicely delineates the pathway under lysosomal stress. The "omics" data serves a valuable resource for the field. More work should be invested into how TAB2/3 are activated at the damaged lysosomes, also to increase novelty in light of previous reports.

Strengths:

Generally, this is a careful and well-conducted study that nicely delineates how the NFkB pathway is activated under lysosomal stress and modulates cell behavior. The "omics" data serves as a valuable resource for the field.

Weaknesses:

While activation of TAB2/3 by K63-linked Ub chains is convincing, more work needs to be done on how they are recruited by distinct damage types to probe relevance for different pathophysiological conditions."

Comments on revisions:

The authors have addressed much of my criticism. Specifically, they have put (with new experiments) the data on the TAB2/3-TAK1 pathway in perspective to the previously reported LUBAC-mediated activation of NFkB. They also addressed the question about the significance of K63-linked chains for TAB2/3 activation with new complementation experiments (a K63-specific NZF mutant failed to rescue).

The third point (types of damage as triggers) raises more questions, though. The authors find that, in contrast to LLOMe, GPN or DC661-induced damage does not activate TAK1 (consistent with lower damage levels). However, the authors still observe K63 ubiquitylation. This goes along with their finding that TAB2 is recruited in the absence of any ubiquitylation (blocked by TAK-243). It argues that TAB2 is recruited by an unknown cue (that may be damage-specific) and then activated by K63. The authors need to clarify whether TAB2 is or is not recruited in the GPN/DC661 conditions (in which K63 occurs, but TAK1 is not activated). The point about the effects of other damage types was also raised by reviewer #1 and should be solved. The fact that TAB2 is recruited independently of K63 should also be visualized in the model. The manuscript will then be an important contribution to the field.

---

## [Author Response]

The following is the authors’ response to the original reviews.

**Reviewer #1 (Public review):**
Summary:Lysosomal damage is commonly found in many diseases including normal aging and age-related disease. However, the transcriptional programs activated by lysosomal damage have not been thoroughly characterized. This study aimed to investigate lysosome damage-induced major transcriptional responses and the underlying signaling basis. The authors have convincingly shown that lysosomal damage activates a ubiquitination-dependent signaling axis involving TAB, TAK1, and IKK, which culminates in the activation of NF-kB and subsequent transcriptional upregulation of pro-inflammatory genes and pro-survival genes. Overall, the major aims of this study were successfully achieved.Strengths:This study is well-conceived and strictly executed, leading to clear and well-supported conclusions. Through unbiased transcriptomics and proteomics screens, the authors identified NF-kB as a major transcriptional program activated upon lysosome damage. TAK1 activation by lysosome damage-induced ubiquitination was found to be essential for NF-kB activation and MAP kinase signaling. The transcriptional and proteomic changes were shown to be largely driven by TAK1 signaling. Finally, the TAK1-IKK signaling was shown to provide resistance to apoptosis during lysosomal damage response. The main signaling axis of this pathway was convincingly demonstrated.Weaknesses:One weakness was the claim of K63-linked ubiquitination in lysosomal damage-induced NF-kB activation. While it was clear that K63 ubiquitin chains were present on damaged lysosomes, no evidence was shown in the current study to demonstrate the specific requirement of K63 ubiquitin chains in the signaling axis being studied. Clarifying the roles of K63-linked versus other types of ubiquitin chains in lysosomal damage-induced NF-kB activation may improve the mechanistic insights and overall impact of this study.Another weakness was that the main conclusions of this study were all dependent on an artificial lysosomal damage agent. It will be beneficial to confirm key findings in other contexts involving lysosomal damage.

We would like to thank Reviewer #1 for the positive and constructive comments on our study. For a main concern regarding the molecular mechanism by which TAB proteins are activated in response to lysosomal damage, we have added the experimental results to support that the lysosomal accumulation of K63 ubiquitin chains serves as a trigger to activate the TAB-TAK1 pathway. We also investigated and discussed the role of LUBAC-mediated M1 ubiquitin chains in NF-kB activation and the effects of other lysosomal-damaging compounds. Please see the response to “Reviewer #3 (Public review): Suggestions:”.

**Reviewer #2 (Public review):**
Summary:Endo et al. investigate the novel role of ubiquitin response upon lysosomal damage in activating cellular signaling for cell survival. The authors provide a comprehensive transcriptome and proteome analysis of aging-related cells experiencing lysosomal damage, identifying transcription factors involved in transcriptome and proteome remodeling with a focus on the NF-κB signaling pathway. They further characterized the K63-ubiquitin-TAB-TAK1-NF-κB signaling axis in controlling gene expression, inflammatory responses, and apoptotic processes.Strengths:In the aging-related model, the authors provide a comprehensive transcriptome and characterize the K63-ubiquitin-TAB-TAK1-NF-κB signaling axis. Through compelling experiments and advanced tools, they elucidate its critical role in controlling gene expression, inflammatory responses, and apoptotic processes.Weaknesses:The study lacks deeper connections with previous research, particularly:• The established role of TAB-TAK1 in AMPK activation during lysosomal damage• The potential significance of TBK1 in NF-κB signaling pathways

We would like to thank Reviewer #2 for the helpful comments on our study. To achieve a more comprehensive understanding of the signaling pathways involved in the lysosomal damage response, we investigated additional related signal mediators, such as TBK1 and LUBAC. The citations related to AMPK have been incorporated.

**Reviewer #3 (Public review):**
Summary:The response to lysosomal damage is a fast-moving and timely field. Besides repair and degradation pathways, increasing interest has been focusing on damaged-induced signaling. The authors conducted both transcriptomics and proteomics to characterize the cellular response to lysosomal damage. They identify a signaling pathway leading to activation of NFkappaB. Based on this and supported by Western blot and microscopy data, the authors nicely show that TAB2/3 and TAK1 are activated at damaged lysosomes and kick off the pathway to alter gene expression, which induces cytokines and protect from cell death. TAB2/3 activation is proposed to occur through K63 ubiquitin chain formation. Generally, this is a careful and well conducted study that nicely delineates the pathway under lysosomal stress. The "omics" data serves as a valuable resource for the field. More work should be invested into how TAB2/3 are activated at the damaged lysosomes, also to increase novelty in light of previous reports.Strengths:Generally, this is a careful and well-conducted study that nicely delineates the pathway under lysosomal stress. The "omics" data serves as a valuable resource for the field.Weaknesses:More work should be invested into how TAB2/3 are activated at the damaged lysosomes, also to increase novelty in light of previous reports. Moreover, different damage types should be tested to probe relevance for different pathophysiological conditions.

We would like to thank Reviewer #3 for the valuable comments on our study. We have added the experimental results to address two concerns raised by Reviewer #3. Please see the response to “Reviewer #3 (Public review): Suggestions:”.

Suggestions:(1) A recent paper claims that NFkappaB is activated by Otulin/M1 chains upon lysosome damage through TBK1 (PMID: 39744815). In contrast, Endo et al. nicely show that ubiquitylation is needed (shown by TAK-243) for NFkB activation but only have correlative data to link it specifically to K63 chains. On page 15, line 11, the authors even argue a "potential" involvement of K63. This point should be better dealt with. Can the authors specifically block K63 formation? K63R overexpression or swapping would be one way. Is the K63 ligase ITCH involved (PMID: 38503285) or any other NEDD4-like ligase? This could be compared to LUBAC inhibition. Also, the point needs to be dealt with more controversially in the discussion as these are alternative claims (M1 vs K63, TAB vs TBK1).

It is well-characterized that the NZF domain of TAB proteins preferentially associates with K63-linked ubiquitin chains. Therefore, we performed the add-back experiment using siRNA-resistant TAB2 WT and mutants incapable of binding to K63-linked ubiquitin chains, dNZF and E685A, to elucidate the requirement of K63 ubiquitin chains for TAK1 activation. We investigated whether the add-back of TAB2 mutants rescues the activation of TAK1 in TAB2-depleted cells (Fig. 2E). TAB2 WT, but not dNZF and E685A, rescued TAK1 activation in response to LLOMe, suggesting that the specific interaction of TAB proteins and K63 ubiquitin chains is a key mechanism to activate TAK1. We also found that the treatment of an E1 inhibitor TAK-243 effectively prevented the lysosomal accumulation of K63 ubiquitin chains, but TAB2 was recruited to damaged lysosomes (Fig. S2B). This suggests that the recruitment of TAB proteins to damaged lysosomes is independent of the association with K63 ubiquitin chains. Collectively, it is postulated that TAB proteins require interaction with K63 ubiquitin chains for TAK1 activation, but not for recruitment to damaged lysosomes. We have added the sentences (p9, lines 7-20, and p10, lines 8-10).

Next, we confirmed that LUBAC functions are essential for NF-kB activation in the lysosomal damage response. RNF31/HOIP is a component of LUBAC that catalyzes M1 ubiquitination. The depletion of RNF31 showed no significant effects on TAK1 activation, but abolished IKK activation (Fig. S4G). It is well-characterized that LUBAC-mediated M1 ubiquitin chains recruit IKK subunits and transduce the signaling to downstream in the canonical pathway. We assume that K63 ubiquitin chains in damaged lysosomes initially activate TAB-TAK1 and trigger LUBAC-mediated M1 ubiquitination, and subsequently, M1 ubiquitination functions to recruit the IKK complex. Consequently, activated TAK1 phosphorylates IKK subunits in damaged lysosomes, leading to NF-kB activation. We also examined whether TBK1 is involved in the activation of NF-kB. TBK1 was phosphorylated upon LLOMe, and depletion of TAB and TAK1 resulted in a slight reduction of TBK1 phosphorylation (Fig. S4D, E). The treatment of a TBK1 inhibitor BX-795 exhibited no or little effects on TAK1 activation, but abolished phosphorylation of IKK and IkBa (Fig. S4F). These suggest that TBK1 is required for the activation of NF-kB. We have added the sentences (p13, line 13-p14, line 10).

As mentioned by Reviewer #3, it is important to identify the E3 ligase responsible for K63 ubiquitination in the lysosomal damage response. We have been aiming to identify such E3 ligase(s). However, depletions of ITCH and other E3 ligases that have been tested exhibited no or little effects on K63 ubiquitination and TAK1 activation. We would like to explore E3 ligase(s) in future study.

(2) It would be interesting to know what the trigger is that induces the pathway. Lipid perturbation by LLOMe is a good model, but does activation also occur with GPN (osmotic swelling) or lipid peroxidation (oxidative stress) that may be more broadly relevant in a pathophysiological way? Moreover, what damage threshold is needed? Does loss of protons suffice? Can activation be induced with a Ca2+ agonist in the absence of damage?

To further clarify the initial trigger that induces TAB-TAK1 activation coupled with lysosomal damage, we examined other damage sources, GPN and DC661, which induce hyperosmotic stress and lipid peroxidation in lysosomes, respectively, thereby resulting in lysosomal membrane damage. Under our experimental conditions, the treatment of these compounds did not result in significant accumulation of Gal-3, indicating a reduced level of lysosomal membrane permeabilization compared with LLOMe (Fig. S2C, D), and no or little TAK1 activation was observed (Fig. S2E). TAB proteins require their association with K63 ubiquitin chains for TAK1 activation. It is therefore postulated that the severe lysosomal membrane permeabilization that triggers the formation and cytosolic exposure of K63 ubiquitin chains may be a determinant of TAB-TAK1 activation. In our future work, we would like to examine broad stimulation of lysosomal damage and further elucidate the initial mechanism of TAB-TAK1 activation. We have added the sentences (p9, line 21-p10, line 7).

(3) The authors nicely define JNK and p38 activation. This should be emphasized more, possibly also in the abstract, as it may contribute to the claim of increased survival fitness.

We further tested whether the inhibition of JNK affects the anti-apoptotic effect (Fig. S5B). The inhibition of JNK resulted in an increase in the cleaved caspase-3. This suggests that the anti-apoptotic action in the lysosomal damage response requires JNK as well as IKK. We have added the sentences in results to emphasize the pivotal role of stress-induced MAPKs (p15, lines 7-11).

**Reviewer #1 (Recommendations for the authors):**
(1) Although the ubiquitination-TAB-TAK1-IKK axis was previously characterized in other contexts, specific evidence supporting lysosomal recruitment of these components by ubiquitination during lysosome damage would be beneficial.

We found that the treatment of an E1 inhibitor TAK-243 abolished the lysosomal accumulation of K63 ubiquitin chains, but TAB2 and TAK1 were recruited to damaged lysosomes (Fig. S2B). This suggests that the recruitment of TAB proteins to damaged lysosomes is independent of the association with K63-linked ubiquitin chains. Next, we investigated whether the add-back of TAB2 mutants incapable of binding K63 ubiquitin chains rescues the activation of TAK1 in TAB2-depleted cells (Fig. 2E). K63 ubiquitin binding of TAB2 was essential for TAK1 activation in response to LLOMe. Taken together, it is suggested that TAB proteins require their interaction with K63 ubiquitin chains for TAK1 activation, but not for recruitment to damaged lysosomes. We have added the sentences (p9, lines 7-20, and p10, lines 8-10). Please also see the response to “Reviewer #3 (Public review): Suggestions:”.

(2) The activation of p38 and JNK by lysosomal damage does not fit well into the main conclusions of the paper, since IKK knockdown was sufficient to block cellular resistance to apoptosis (caspase cleavage in Fig. 5f). Are p38 and JNK also important for cell survival during lysosomal damage?

We found that the inhibition of JNK resulted in an increase in the cleaved caspase-3, suggesting that the anti-apoptotic action in the lysosomal damage response requires both IKK and JNK (Fig. S5B). We have added the sentences (p15, lines 7-11).

(3) Cell death tests are recommended to support the conclusions related to apoptosis.

As suggested by Reviewer #1, we performed the cell death assay using propidium iodide (PI) and confirmed that HeLa cells co-treated with LLOMe and TAK-243 or HS-276 exhibited increased cell death (Fig. 5E). This indicates a direct correlation between the degree of caspase-3 cleavage and cell death, possibly apoptosis.

(4) Page 8, line 19-21, gal3 is not exposed upon lysosomal damage. It is recruited from the cytosol by the exposed beta-galactoside-containing glycans on lysosomal membrane proteins.

We have corrected the corresponding sentence (p7, lines 17-20).

(5) Carefully checking grammar throughout the text is recommended. Below are a few examples:a) Page 4, line 10, remove "that".b) "K63 ubiquitin" shall be replaced with "K63 ubiquitination" or "K63 ubiquitin chains".c) Page 8, line 9, "remain" should be "remains".

We have carefully checked the revised manuscript.

**Reviewer #2 (Recommendations for the authors):**
Despite the novelty and significance of these findings in advancing the field, several technical and experimental limitations require further clarification:

We have responded to each comment. Please see below.

The manuscript should introduce or discuss previous research showing that TAB-TAK1 facilitates AMPK activation during lysosomal damage and TAK1's increased association with damaged lysosomes (PMID: 31995728).

We have added the reference (PMID: 31995728) and the sentences (p17, lines 15-20).

Figure 2A: The differential LAMP1 staining intensity between control and LLOMe-treated cells needs explanation. The weaker LAMP1 signal in control and puncta changes, especially during 5-minute LLOMe treatment, require detailed clarification

We have added the explanation (p8, lines 17-21).

Recent literature (PMID: 34585663) reports TBK1 activation during lysosomal damage. The authors should investigate or discuss whether TBK1 potentially contributes to NF-κB signaling in this context.

We experimentally investigated whether TBK1 is involved in the TAB-TAK1 pathway. We confirmed that TBK1 was activated upon LLOMe (Fig. S4D). Depletions of TAB and TAK1 exhibited a modest decrease in TBK1 phosphorylation (Fig. S4E). The inhibition of TBK1 by BX-795 did not affect TAK1 activation, but abolished phosphorylation of IKK and IkBa (Fig. S4F). This suggests that TBK1 is required for NF-kB activation. We have added the reference (PMID: 34585663) and the sentences (p13, lines 13-21, p14, lines 8-10, and p18, lines 15-20).

The introduction of lysosomal damage response lacks comprehensive mechanistic information. For example, while ESCRT is discussed, other critical mechanisms such as lipid transfer and stress granule formation in lysosomal repair should be incorporated. Moreover, mTOR and AMPK signaling pathways undergo significant changes upon lysosomal damage.

We have added the sentences (p3, lines 16-18, and p3, line 21-p4, line 1).

The statement "lysosomal permeabilization causes the dissociation of mTORC1 from lysosomes" should explicitly reference PMID: 29625033.

We have added the suggested reference (PMID: 29625033, p4, line 19).

The claim that "The elimination of damaged lysosomes through lysophagy requires a period of more than half a day" needs a specific publication citation.

We have added the reference (PMID: 23921551) to claim the time-scale of lysosomal clearance (p4, line 21).

Figure 1G: The label "WO after 2h" lacks explanation in the figure legend and requires detailed interpretation.

To simplify the figures, we have deleted the label “WO after 2 h” (Fig. 1G, 3F, 5D, F-J, S4G, S5A). Instead, we have added the explanation in the figure legends (Fig. 1G).

**Reviewer #3 (Recommendations for the authors):**
(1) page 8, line 13: it is recommended to phrase colocalisation "at" damaged lysosomes rather than "in" damaged lysosomes as the resolution does not allow the claim of influx into lysosomes.

We have corrected the word (p8, line 17).

(2) page 11, line 22: why is "whereas" used to link two events driven by the same mechanism.

We have corrected the word (p13, line 8).